# A general mechanism for initiating the bacterial general stress response

**Rishika Baral[1,2], Kristin Ho[1], Ramasamy P Kumar[1], Jesse B Hopkins[3], Maxwell B Watkins[4], Salvatore LaRussa[1,2], Suhaily Caban-Penix[1,5], Logan A Calderone[1,2], Niels Bradshaw[1]***

[1]Department of Biochemistry, Brandeis University, Waltham, United States; [2]Graduate program in Biochemistry and Biophysics, Brandeis University, Waltham, United States; [3]Biophysics Collaborative Access Team (BioCAT), Department of Physics, Illinois Institute of Technology, Chicago, United States; [4]Biophysics Collaborative Access Team (BioCAT), Department of Biology, Illinois Institute of Technology, Chicago, United States; [5]Graduate program in Molecular and Cell Biology, Brandeis University, Waltham, United States

*For correspondence:
niels@brandeis.edu

Competing interest: The authors declare that no competing interests exist.

## eLife Assessment

This **important** study combines genetic analysis, biochemistry, and structural modeling to reveal new insights into how changes in protein-protein structure activate signal transduction as part of the bacterial general stress response. The data, which was collected using validated and standard methods, and its interpretations are **convincing**; however, to fully meet the title's promise, additional experimental evidence is needed to strengthen the proposed model and its potential application to other systems. This manuscript will be of broad interest to microbiologists, structural biologists, and cell biologists.

**Abstract** The General Stress Response promotes survival of bacteria in adverse conditions, but how sensor proteins transduce species-specific signals to initiate the response is not known. The serine/threonine phosphatase RsbU initiates the General Stress Response in *Bacillus subtilis* upon binding a partner protein (RsbT) that is released from sequestration by environmental stresses. We report that RsbT activates RsbU by inducing otherwise flexible linkers of RsbU to form a short coiled-coil that dimerizes and activates the phosphatase domains. Importantly, we present evidence that related coiled-coil linkers and phosphatase dimers transduce signals from diverse sensor domains to control the General Stress Response and other signaling across bacterial phyla. This coiled-coil linker transduction mechanism additionally suggests a resolution to the mystery of how shared sensory domains control serine/threonine phosphatases, diguanylate cyclases and histidine kinases. We propose that this provides bacteria with a modularly exchangeable toolkit for the evolution of diverse signaling pathways.

## Introduction

The General Stress Response (GSR) allows bacteria to survive in hostile and changing environments (*Bergkessel et al., 2016*; *Harms et al., 2016*; *Gottesman, 2019*; *Guldimann et al., 2016*; *Hecker et al., 2007*). A defining feature of the GSR is that it controls transcription of a large and diverse set of genes, dramatically reshaping the physiology of the organism (*Gottesman, 2019*; *Guerreiro et al., 2020*; *Boekhoud et al., 2020*; *Boylan et al., 1993*). Another defining feature is that the GSR is induced by a variety of stress conditions that are frequently associated with each other, including

**Figure 1.** The linker controls RsbU activation through a conserved switch element. (**A**) Reaction scheme of the RsbU-dependent pathway of the environmental stress response. RsbU is activated by its partner protein RsbT (T, purple) to dephosphorylate its substrate protein RsbV-P (**V–P**). This initiates transcription by displacing σ$^B$ from inhibition by the anti-sigma factor RsbW (**W**). (**B**) Amino acid substitutions in RsbU (Q94L and M166V) cause enhanced σ$^B$ activity in *B. subtilis*. Strains carrying a lacZ reporter of σ$^B$ activity (*ctc-lacZ*) were plated on X-gal indicator plates in the presence (top) or absence (bottom) of IPTG to induce expression of the *rsbU* constructs. Strains were deleted for *rsbT* (–) or not (+). A strain overexpressing *rsbT* is indicated by oe. Plates were imaged after 24 h of growth at 37 °C, which was shorter than the time used to visualize *lacZ* expression from the *rsbU$^{M166V}$* strain previously (*Ho and Bradshaw, 2021*). (**C**) Domain diagram of RsbU. The N-terminal domain of RsbU (blue; amino acids 1–81) is joined to the α-helical linker domain (grey; 82-112), which connects it to the PPM phosphatase domain (grey box; 121-335). The regulatory switch element of the PPM phosphatase domain is colored orange (156-201). The location of RsbT-bypass mutations are highlighted in green. (**D**) AlphaFold2 model of RsbU dimer. Domains are colored as in (**C**) and the Cα atoms of residues Q94 and M166 are shown as spheres.

The online version of this article includes the following figure supplement(s) for figure 1:

**Figure supplement 1.** PAE plots of AlphaFold2 structure predictions.

**Figure supplement 2.** AlphaFold2 structure predictions with mapped pLDDT scores.

**Figure supplement 3.** Q94L stabilizes the packing interface of the coiled-coil of the linker.

stationary phase, nutrient depletion, cellular damage, and (for pathogens) host defense mechanisms (*Gottesman, 2019*). This multiplicity of initiating conditions allows cells to anticipate and prepare for accumulating adversity, suggesting why the GSR is present in every bacterial species that has so far been analyzed. The GSR additionally contributes to pathogenesis through expression of genes controlling virulence, biofilm formation, and quorum sensing (*Kint et al., 2017*; *Battesti et al., 2015*; *Bartolini et al., 2019*). Inactivation of the GSR, in turn, promotes the transition to long-term slow growth and antibiotic resistance, which are particularly important for persistent infections (*Tuchscherr and Löffler, 2016*; *Herbert et al., 2010*; *Bergkessel et al., 2016*). Thus, deciphering the mechanisms that govern GSR activity is critical to understanding how bacteria interact with and adapt to their environments.

A major barrier to understanding initiation of the GSR is that we lack a unique, generalizable mechanistic model for how the activating signaling proteins have evolved to respond to species-specific signals. In a widespread mechanism for initiating the GSR, a member of the PPM family of serine/threonine phosphatases dephosphorylates a phosphorylated substrate protein to activate an alternative σ-factor through a 'partner-switching' mechanism (*Hecker et al., 2007*), but how the phosphatase receives and is activated by an upstream signal has not been determined.

In this study, we address how the GSR phosphatase RsbU from *Bacillus subtilis* is regulated (*Yang et al., 1996*) and present evidence that this mechanism is widely conserved. RsbU dephosphorylates a single phospho-serine on its substrate protein (RsbV) to release the GSR transcription factor (σ$^B$) from inhibition by an anti-sigma factor (RsbW; *Figure 1A*). RsbU has an N-terminal four-helix bundle domain that dimerizes RsbU and is also the binding site for RsbT, which activates RsbU as a phosphatase (*Figure 1C and D*; *Delumeau et al., 2004*). RsbT is sequestered in a megadalton stress sensing

complex called the stressosome, and is released to bind RsbU in response to specific stress signals including ethanol, heat, acid, salt, and blue light (*Hecker et al., 2007*; *Marles-Wright et al., 2008*). We and others have reconstituted RsbU activation by RsbT (*Delumeau et al., 2004*; *Ho and Bradshaw, 2021*), but how RsbT activates RsbU was not known.

This 'partner-switching' mechanism is broadly conserved, and the activating phosphatases are observed to have diverse N-terminal sensory domains including PAS (*Vijay et al., 2000*; *Greenstein et al., 2009*; *Martínez et al., 2009*), response-receiver (*de Been et al., 2010*; *Rodríguez-Martínez et al., 2023*), GAF (*Kint et al., 2019*), and HAMP (*Mittenhuber, 2002*). These signaling domains are frequently found as N-terminal regulators of histidine kinases and GGDEF diguanylate cyclases in bacteria, but whether or not there are shared mechanistic features between the regulatory principles of PPM phosphatases and these other signaling effector families is not known (*Möglich et al., 2009*; *Galperin, 2006*).

We previously described a mechanism for allosteric control of the related *B. subtilis* phosphatase SpoIIE, which regulates endospore formation through a partner-switching mechanism analogous to RsbU control of the GSR (*Ho and Bradshaw, 2021*; *Bradshaw et al., 2017*). Two helices of the PPM domain (α1 and α2), which we refer to as the switch element, change conformation to recruit metal cofactor and activate SpoIIE (*Bradshaw et al., 2017*). Based on analysis of structural and genetic data, we conjectured that this conformational change exemplifies a common mechanism among diverse phosphatases. However, we could not propose a specific model for how unrelated regulatory domains modularly control the switch conformation. For RsbU, we found that RsbT enhances phosphatase activity by increasing the efficacy of metal cofactor binding and that substitution of an amino acid in α1 (RsbU$^{M166V/L}$) activates RsbU in the absence of RsbT (*Ho and Bradshaw, 2021*). These results supported our proposal that the switch element controls RsbU activation, but they did not reveal how RsbT binding to the N-terminus of RsbU is transduced to the switch.

One broadly conserved feature of GSR phosphatases (absent in SpoIIE) is an α-helical linker of variable length, which is N-terminal to the PPM phosphatase domain. Genetic data suggest that this linker regulates a paralogous *B. subtilis* GSR phosphatase (RsbP; *Brody et al., 2009*). Here, we report that binding of RsbT to the N-terminus of RsbU rigidifies an otherwise flexible linker to dimerize the C-terminal phosphatase domains through an interface mediated by the switch element. Importantly, and as we now report, this mechanism is generalizable to other PPM phosphatases and reveals a widespread feature of bacterial signaling.

## Results

### The linker controls RsbU activation through a predicted phosphatase dimer

We previously conducted a genetic screen (*Ho and Bradshaw, 2021*) to identify features of RsbU that are important for phosphatase regulation by isolating gain-of-function variants that are active in the absence of RsbT. Focusing on the phosphatase domain in our initial study, we reported identification of an amino acid substitution in α1 (RsbU$^{M166V/L}$) that links the switch element to RsbU activation (*Ho and Bradshaw, 2021*). From this screen, we isolated a second variant, RsbU$^{Q94L}$, located in the linker of RsbU between the N-terminal dimerization domain and the C-terminal phosphatase domain (*Figure 1B–D*). *rsbU*$^{Q94L}$ strains activated σ$^B$ in the absence of *rsbT*. This activation is comparable to when *rsbT* is overexpressed to exceed the capacity of the stressosome and is stronger than what we observe in strains expressing *rsbU*$^{M166V/L}$ (*Figure 1B*).

To visualize how the Q94L and M166V/L substitutions could impact RsbU structure and activity, we generated an AlphaFold2 model of an RsbU dimer (*Jumper et al., 2021*; *Mirdita et al., 2022*; *Figure 1D*). The prediction of the complex was of high confidence (*Figure 1—figure supplements 1–2*) and had a dimer of the N-terminal region consistent with a previous experimental structure (*Delumeau et al., 2004*). In our model, the linkers form extended alpha-helices (82-112) that cross to connect dimers of the RsbU N-terminal domains and C-terminal phosphatase domains, with Q94 buried in the linker interface and M166 buried in the core of the phosphatase domain (*Figure 1D*).

The crossing linker helices have similar conformation to homologous linkers observed in crystal structures of *Pseudomonas aeruginosa* RssB, a phosphatase/adapter protein that also controls the GSR (PDB 3F7A and 3EQ2, *Figure 1—figure supplement 3*). RssB has an N-terminal response-receiver

domain in place of the RsbU four helix bundle dimerization domain, suggesting that linker conformation is a conserved feature of PPM phosphatase dimers with divergent N-terminal regulatory domains (*Figure 1—figure supplement 3*). The linker helices in the RsbU prediction and the RssB structure splay apart as they enter the PPM domain with similar angle (pitch angles (α) based on Crick parameters for coiled-coils –17.6° and –18.4°, respectively) and pack against α–1 of the PPM fold (*Grigoryan and Degrado, 2011*). Interestingly, we observed that other bacterial signaling proteins that are regulated by coiled-coil linkers had similar pitch angle for the helices connecting to the catalytic domains (for example the dimeric histidine kinase CheA, 1B3Q, α –16.8°, and the dimeric GGDEF diguanylate cyclase PadC, 5LLW, α –14.3°; *Bilwes et al., 1999*; *Gourinchas et al., 2017*; *Grigoryan and Degrado, 2011*; *Figure 1—figure supplement 3*). Thus, the linker of RsbU could be a conserved regulatory feature, analogous to linkers of histidine kinases and GGDEF diguanylate cyclases.

Analysis of the RsbU dimer model using Socket2 identifies amino acids 83–97 as a two-turn coiled coil, with Q94 unfavorably packing against leucine 90 (*Kumar and Woolfson, 2021*; *Figure 1—figure supplement 3*). To test whether hydrophobicity at position 94 promotes an active state, we tested the activity of RsbU alleles with different amino acid substitutions at position 94 for $\sigma^B$ in wildtype and *rsbT* deleted strains. We found that substitutions of Q94 with amino acids that could pack at the intersection of the crossing linker helices (hydrophobic residues isoleucine, valine, and methionine, as well as tyrosine) resulted in mild, RsbT-independent activation of $\sigma^B$ that was further heightened by RsbT (*Figure 1—figure supplement 3*). Aromatic hydrophobic residues (phenylalanine and tryptophan) drove constitutive activation of RsbU but did not bypass the requirement for *rsbT* on the chromosome (*Figure 1—figure supplement 1*). Substitutions of small or charged residues (alanine, glycine, glutamate, and asparagine) did not lead to activity in either strain (*Figure 1—figure supplement 3*). These results are consistent with a model in which burial of the amino acid at position 94, such as in the crossed helices of our prediction for RsbU dimer structure, drives RsbU activation.

## RsbT binding to RsbU is influenced by the linker

One prediction of our hypothesis that RsbT stabilizes the crossed alpha helices of the RsbU dimer, is that RsbT should bind more tightly to RsbU$^{Q94L}$ than to RsbU because the coiled-coil conformation that RsbT binds would be more energetically favorable. To test this, we measured the binding affinity of RsbT for RsbU using fluorescence anisotropy. From these data we obtained a $K_D$ of 4.5±1 µM, which decreased approximately 10-fold for RsbU$^{Q94L}$ (0.56±0.2 µM; *Figure 2A*). Thus, RsbU$^{Q94L}$ stimulates RsbU phosphatase activity by an on-pathway mechanism, supporting the idea that RsbT binding stabilizes the linker in a dimerized conformation to activate RsbU.

## RsbU phosphatase activation is controlled by the linker

We previously found that RsbT activates RsbU by simultaneously decreasing the $K_M$ for manganese cofactor and increasing the $k_{cat}$ for dephosphorylation of RsbV-P (*Ho and Bradshaw, 2021*). To test whether RsbU Q94L substitution works by the same kinetic mechanism, we measured dephosphorylation of $^{32}P$-phosphorylated RsbV by RsbU$^{Q94L}$ (*Figure 2B*). The $k_{cat}$ of RsbU$^{Q94L}$ was nearly the same as for the basal activity of RsbU (2.4±0.1 min$^{-1}$ compared to 1.4±0.1 min$^{-1}$, and stimulated more than 10-fold by RsbT, 15±0.4 min$^{-1}$; *Figure 2B*; *Ho and Bradshaw, 2021*). However, RsbU$^{Q94L}$ had a $K_M$ for MnCl$_2$ (1.4±0.2 mM) that is more than 50-fold reduced compared to RsbU alone (77±9 mM; *Ho and Bradshaw, 2021*), and similar to the $K_M^{MnCl2}$ of RsbU in the presence of RsbT (0.98±0.07 mM *Ho and Bradshaw, 2021*; *Figure 2B*). Addition of RsbT did not change the $K_M$ of RsbU$^{Q94L}$ for MnCl$_2$ (1.9±0.6 mM), but the maximum catalytic rate of dephosphorylation increased nearly four-fold to 9.0±0.64 min$^{-1}$, approaching the rate of RsbU stimulated by RsbT (*Figure 2B*). Thus, RsbU$^{Q94L}$ substantially recapitulates RsbT activation by decreasing the $K_M$ for Mn$^{2+}$, demonstrating that the linker transduces RsbT binding to control the active site of RsbU (*Figure 2C*). This is consistent with previous findings that the allosteric switch controlling phosphatase activity of SpoIIE couples dimerization to cofactor interaction.

## RsbT binds to the linker and N-terminal domain

To visualize how RsbT could control the linker to activate RsbU, we generated an AlphaFold2 prediction of a heterotetrameric complex between RsbT and RsbU (*Figure 3A*). In this model, RsbU adopts a conformation very similar to our prediction of the RsbU dimer (RMSD 1.36 Å), suggesting that

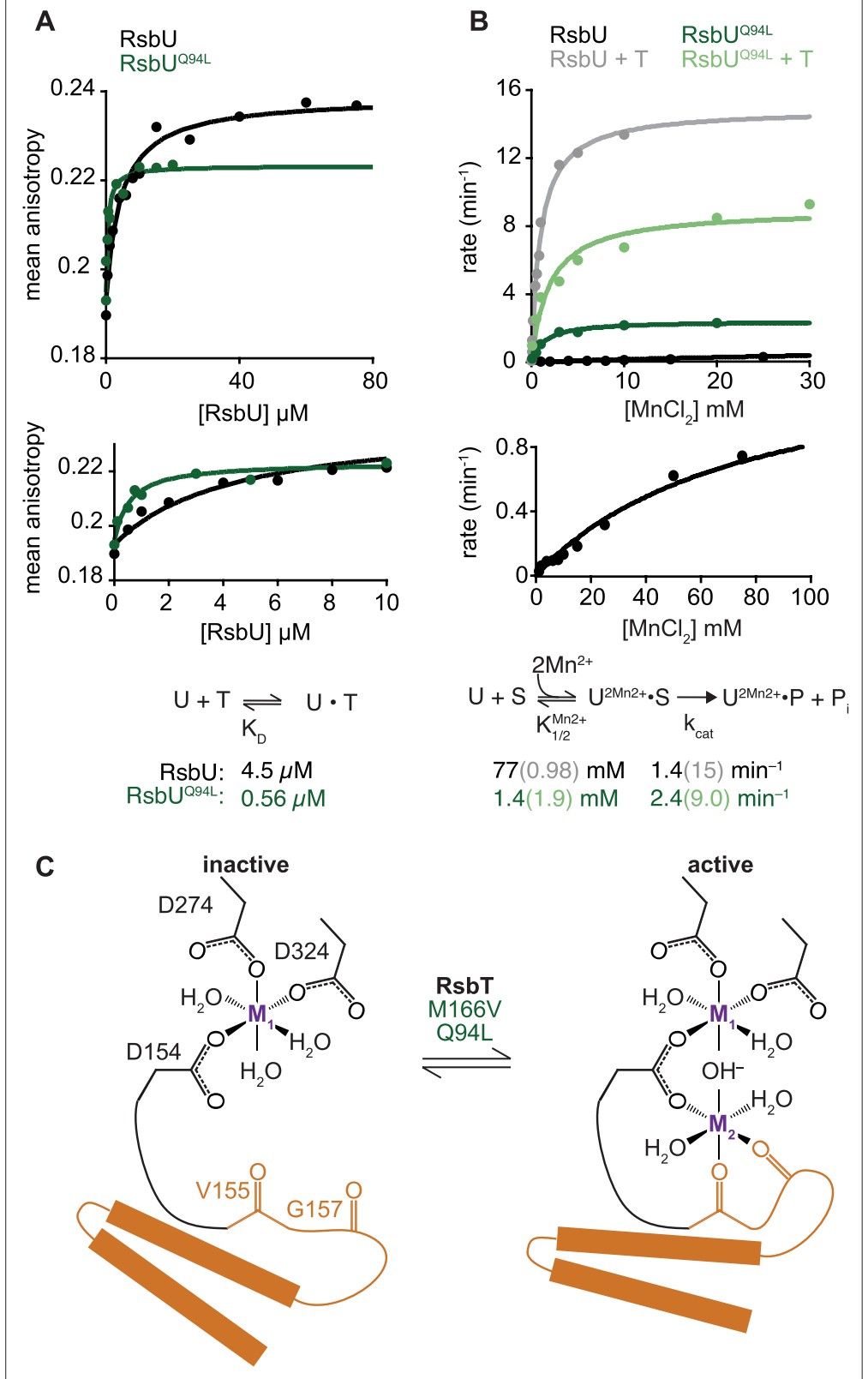

**Figure 2.** RsbU linker influences RsbT binding and phosphatase activity. (**A**) Mean anisotropy of RsbT-TMR (200 nM) is plotted as a function of RsbU concentration. The curves are fits to a quadratic binding equation using non-linear curve fitting (RsbU (black) $K_D$ = 4.5 ± 0.96 µM, RsbU$^{Q94L}$ (green) $K_D$ = 0.56 ± 0.15 µM). The lower plot shows the same data plotted to display RsbU concentrations to 10 µM. Below is the reaction scheme of RsbT binding to RsbU

*Figure 2 continued on next page*

*Figure 2 continued*

with the values for the calculated $K_D$ below. (**B**) The rate of dephosphorylation of RsbV-P is plotted as a function of concentration of $MnCl_2$ for RsbU (black) and $RsbU^{Q94L}$ (green) in the presence (dark) and absence (light) of RsbT. Curves are fits to the Michaelis-Menten equation using non-linear curve fitting. The $k_{cat}$ of wild-type RsbU is $15\pm0.35$ min$^{-1}$ with RsbT and $1.4\pm0.077$ min$^{-1}$ without RsbT and the $k_{cat}$ of $RsbU^{Q94L}$ is $9.0\pm0.64$ min$^{-1}$ with RsbT and $2.4\pm0.084$ min$^{-1}$ without RsbT. The $K_M^{MnCl2}$ of wild-type RsbU is $0.98\pm0.069$ mM with RsbT and $77\pm9.0$ mM without RsbT and the $K_M^{MnCl2}$ of $RsbU^{Q94L}$ is $1.4\pm0.19$ mM with RsbT and $1.9\pm0.56$ mM without RsbT. The lower plot shows the data for wild-type RsbU in the absence of RsbT shown in the upper plot including higher $MnCl_2$ concentrations and with rescaled Y-axis. Below is a summary of a reaction scheme of RsbU dephosphorylating RsbV-P (denoted as S) with the $K_M^{MnCl2}$ and $k_{cat}$ values below. Data for RsbU are reproduced from *Ho and Bradshaw, 2021*. (**C**) A cartoon model of how the switch helices rotate to activate RsbU. Binding of RsbT or mutation moves the switch helices into place during activation to coordinate metal M2. The residues of RsbU that are hypothesized to coordinate metals are shown as sticks (based on homology to RsbX).

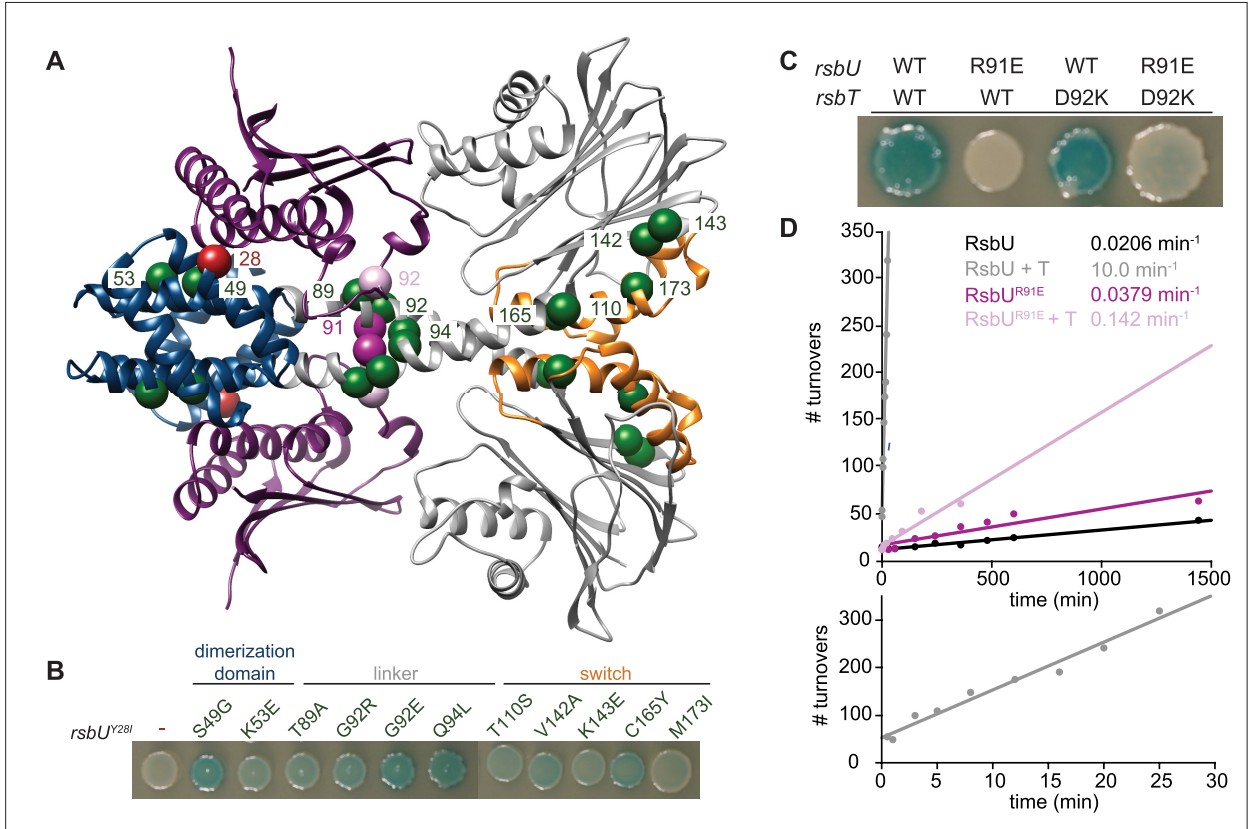

**Figure 3.** RsbT has a hybrid interface with RsbU N-terminal dimerization domain and linker. (**A**) An AlphaFold2 model of a heterotetrameric 2RsbT/2RsbU complex is shown in ribbon representation. RsbT (purple) binds with an interface that spans the N-terminal domains (blue) and α-helical linkers (grey) of both RsbU protomers. The PPM phosphatase domains are colored grey and the regulatory switch elements (α1 and α2) are colored orange. The Cα positions of RsbU$^{Y28}$ (red), *rsbU*$^{Y28I}$ suppressor substitutions (green), RsbU$^{R91}$ (dark pink), and RsbT$^{D92}$ (pink) are shown as spheres with the residue numbers indicated. (**B**) Strains with a σ$^B$ LacZ reporter were plated on IPTG/X-gal plates to induce expression of plasmid borne *rsbT* and *rsbU*$^{Y28I}$ and visualize σ$^B$ activity (indicated by blue pigmentation after 24 hr of growth at 37 °C). Additional amino acid substitutions in the *rsbU* gene are indicated above. (**C**) Strains with indicated changes to the RsbT and RsbU amino acid sequence were plated as in (**B**) and plates were imaged after 36 hr of growth at 37 °C. (**D**) R91E does not decrease the basal phosphatase activity of RsbU. Plots show dephosphorylation of RsbV-P (25 µM) by RsbU (black) and RsbU$^{R91E}$ (pink) (0.5 µM) in the presence of 10 mM $MnCl_2$ and the presence or absence of 10 µM RsbT (light colors). Linear fits of the data are shown, with the slopes indicating the observed rates ($k_{obs}$): RsbU $0.0206\pm0.00124$ min$^{-1}$, RsbU +RsbT $10.0\pm0.628$ min$^{-1}$, RsbU$^{R91E}$ is $0.0379\pm0.00543$ min$^{-1}$, RsbU$^{R91E}$ + RsbT $0.142\pm0.0161$ min$^{-1}$. The lower plot is RsbU in the presence of RsbT to 30 min.

The online version of this article includes the following figure supplement(s) for figure 3:

**Figure supplement 1.** RsbT binds to the linker.

our AlphaFold2 RsbU dimer model represents the active state of RsbU even when RsbT was not included in the prediction (a known feature of AlphaFold2 models *Chakravarty and Porter, 2022*). The predicted structure places RsbT in a hybrid interface that spans the N-terminal dimerization domain of both RsbU monomers and the linker of one monomer (*Figure 3A*). Supporting the validity of the AlphaFold2 2RsbT/2RsbU model, residues previously implicated in RsbT/U binding (*Hardwick et al., 2007*; Y28 and E24 from one RsbU monomer, I74, and I78 from the other RsbU monomer, and RsbT R19, R23, and N24) are all buried in the predicted interface (*Figure 3—figure supplement 1*).

To gain further insight into how RsbT binding is transduced to RsbU linker conformation and phosphatase activity, we performed a genetic screen to isolate suppressors of RsbU$^{Y28I}$, which was previously observed to abolish binding to RsbT (*Hardwick et al., 2007*). After screening a library of PCR mutagenized *rsbU*$^{Y28I}$ and *rsbT*, we identified 11 variants that introduced substitutions to RsbU that suppress *rsbU*$^{Y28I}$: S49G, K53E, T89A, G92E, G92R, Q94L, T110S, V142A, K143E, C165Y, and M173I (*Figure 3A–B*). We additionally isolated revertants that restored Y28 suggesting that the screen was near saturation, but did not isolate any suppressors in *rsbT*, which was also mutagenized.

Mapping the RsbU$^{Y28I}$ suppressor mutations onto our 2RsbU/2RsbT model suggests that they fall into three classes (*Figure 3A and B*): (*1*) Substitutions in the RsbU N-terminal dimerization domain that map near the predicted RsbT binding site (S49G and K53E), supporting the predicted interface. (*2*) Substitutions that map near predicted contacts between RsbT and the RsbU linker (T89A, G92E/R, and Q94L), providing evidence that contacts between RsbT and the linker promote RsbU activation. (*3*) Substitutions that cluster around the regulatory switch element of the RsbU phosphatase domain (helices α1 and α2 of the PPM fold, T110S, V142A, K143E, C165Y, and M173I). We hypothesize that these substitutions rescue RsbU$^{Y28I}$ inactivation by reducing the energetic cost (normally provided by RsbT binding) to transition the RsbU phosphatase domain to the active conformation. Together these genetic data provide independent support for the AlphaFold2 model of the 2RsbT/2RsbU complex, and suggest that RsbT activates RsbU by transmitting signals to the switch element in the RsbU phosphatase domain through a coiled-coil linker.

## Contacts between RsbT and the RsbU linker drive phosphatase activation

To test the model that RsbT activates RsbU by directly interacting with the linker to dimerize the RsbU phosphatase domains, we introduced a charge swap at position R91 that would abolish a predicted salt-bridge with RsbT D92 (*Figure 3C*). RsbU$^{R91E}$ abolished σ$^B$ activity for cells overexpressing *rsbT* (*Figure 3C*), and in biochemical assays RsbT stimulation of RsbU$^{R91E}$ phosphatase activity was abolished (*Figure 3D*). Finally, we introduced a potentially compensatory charge switch at position D92 of RsbT (RsbT$^{D92K}$; *Figure 3C*). Expression of *rsbT*$^{D92K}$ partially restored σ$^B$ activity to *rsbU*$^{R91E}$ expressing cells, consistent with direct interaction between these residues (*Figure 3C*). Together, these results suggest that contacts between RsbT and the linker of RsbU are responsible for phosphatase activation.

## The RsbU linker is flexible in the absence of RsbT

We considered two models for why RsbU is inactive in the absence of RsbT. One possibility is that inactive RsbU resembles the dimeric structure predicted by AlphaFold2 and that RsbT induces a subtle conformational change, for example changing the registry or pitch of the crossing linker helices. A second possibility is that inactive RsbU is dimerized by the N-terminal domains but that the linkers of inactive RsbU are flexible and that the phosphatase domains only interact with each other when RsbT orders the linkers into a crossing conformation. To distinguish between these two models, we performed small angle X-ray scattering coupled to size exclusion chromatography with multi-angle light scattering (SEC-MALS-SAXS) to assess the shape and flexibility of RsbU dimers in solution (*Table 1* and *Figure 4—figure supplements 1–2* S4A). RsbU eluted as a single peak from the size exclusion column with an average molecular weight of 77 kDa (calculated dimeric molecular weight 77.3 kDa) determined by multi-angle light scattering (MALS; *Figure 4—figure supplement 1*). However, our sample had a minor contaminant of heterodimeric complexes of RsbU in which one monomer is C-terminally truncated that could influence the SAXS analysis (*Figure 4—figure supplement 1*). We therefore deconvoluted the SAXS profiles using evolving factor analysis (EFA) in RAW to isolate scattering profiles from the individual components (*Hopkins et al., 2017*; *Meisburger et al., 2016*; *Figure 4—figure supplement 2*).The EFA deconvolution identified a major component (MW 75.6 kDa calculated

**Table 1.** SAXS experimental details.

**(a) Sample details**

| | RsbU SASDU85 | RsbU$^{Q94L}$/RsbT SADSDU95 |
|---|---|---|
| Organism | *B. subtilis* | *B. subtilis* |
| Description: | RsbU (1-335) Uniprot ID: P40399 GPG scar from 6His cleavage | RsbU$^{Q94L}$ (1-335) Uniprot ID: P40399; RsbT (1-133) Uniprot ID: P42411, 6His with 3 C cleavage site |
| Extinction coefficient ε (280 nm) | 26,820 M$^{-1}$cm$^{-1}$ | RsbU$^{Q94L}$: 26,820 M$^{-1}$cm$^{-1}$ RsbT: 13,980 M$^{-1}$cm$^{-1}$ |
| Molecular mass *M* from chemical composition | 38.6 kDa | RsbT: 14.3 kDa RsbU$^{Q94L}$: 38.6 kDa |
| For SEC-SAX, loading volume/ concentration injection volume, flow rate | 3.5 mg ml$^{-1}$ 300 µl, 0.6 ml min$^{-1}$ | 2.5 mg ml$^{-1}$ 300 µl, 0.6 ml min$^{-1}$ |
| Solvent composition and source | 20 mM HEPES pH 7.5, 100 mM NaCl, 5 mM DTT | 20 mM HEPES pH 7.5, 100 mM NaCl, 5 mM DTT |

**(b) SAX data collection parameters**

| | |
|---|---|
| Source, instrument and description | BioCAT facility at the Advanced Photon Source beamline 18ID with Pilatus3 X1 M (Dectris) detector |
| Wavelength | 1.033 Å |
| Beam size (µm$^2$) | 150 (h) x 25 (v) focused at the detector |
| Camera length | 3.682 m |
| *q*-measurement range | 0.0027–0.33 Å$^{-1}$ |
| Absolute scaling method | Glassy Carbon, NIST SRM 3600 |
| Basis for normalization to constant counts | To transmitted intensity by beam-stop counter |
| Method for monitoring radiation damage | Automated frame-by-frame comparison of relevant regions using CORMAP algorithm (*Franke et al., 2015*) implemented in BioXTAS RAW |
| Exposure time | 0.5 s exposure time with a 1 s total exposure period (0.5 s on, 1.5 s off) of entire SEC elution |
| Sample configuration | SEC-MALS-SAXS. Size separation used a Superdex 200 10/300 Increase GL column and a 1260 Infinity II HPLC (Agilent Technologies). UV data was measured in the Agilent, and MALS-DLS-RI data by DAWN HELEOS-II (17 MALS +1 DLS channels) and Optilab T-rEX (RI) instruments (Wyatt Technology). SAXS data was measured in a sheath-flow cell (*Kirby et al., 2016*), effective path length 0.542 mm. |
| Sample temperature | 22 °C |

**(c) Software employed for SAXS data reduction, analysis and interpretation**

| | |
|---|---|
| MALS-DLS-RI analysis | Astra 7 (Wyatt) |
| SAX data reduction | Radial averaging; frame comparison, averaging, and subtraction done using BioXTAS RAW 2.1.4 (*Hopkins et al., 2017*). Deconvolution of overlapping peaks by EFA (*Meisburger et al., 2016*) as implemented in RAW. |
| Basic analyses: | Guinier fit and molecular weight using BioXTAS RAW 2.1.4, P(r) function using GNOM (*Svergun, 1992*). RAW uses MoW and Vc molecular weight methods (*Rambo and Tainer, 2011*; *Piiadov et al., 2019*) ATSAS Version 3.2.1 (*Manalastas-Cantos et al., 2021*). |
| FoXS | Calculated predicted scattering profiles from the model (FoXS) (*Schneidman-Duhovny et al., 2013*; *Schneidman-Duhovny et al., 2016*) |

*Table 1 continued on next page*

*Table 1 continued*

| MultiFoxS | Calculated predicted scattering profiles from the multicomponent fits from a generated flexible structure ensemble (MultiFoXS) (*Schneidman-Duhovny et al., 2016*) | |
|---|---|---|
| **(*d*) Structural parameters** | | |
| ***Figure 4* components** | **c0: RsbU dimer** | **c1: 2T/2U$^{Q94L}$ heterotetramer** |
| Guinier Analysis | | |
| $I(0)$ (Arb.) | 5.632±0.008 | 0.564±0.003 |
| $R_g$ | 39.05±0.16 Å | 35.06±0.43 Å |
| $q$-range | 0.0036–0.0234 Å$^{-1}$ | 0.0033–0.0336 Å$^{-1}$ |
| $q_{max}R_g$ | 0.916 | 1.178 |
| Coefficient of correlation, $r^2$ | 0.995 | 0.88 |
| Volume (adjusted $V_P$ as SAXS MoW2) | 116000 Å$^3$ | 124000 Å$^3$ |
| MW, MoW2 method (ratio to expected) | 96.2 kDa (1.25) | 103.2 kDa (0.98) |
| MW, Vc method (kDa) (ratio to expected) | 75.6 kDa (0.98) | 83.9 kDa (0.80) |
| $P(r)$ analysis | | |
| $I(0)$ (Arb.) | 5.648±0.007 | 0.561±0.003 |
| $R_g$ | 39.98±0.12 Å | 34.87±0.36 Å |
| $D_{max}$ | 168 Å | 176 Å |
| $q$-range | 0.0035–0.3327 Å$^{-1}$ | 0.005–0.3327 Å$^{-1}$ |
| $\chi^2$ (total estimate from GNOM) | 1.257 (0.729) | 1.374 (0.691) |
| FoXS | | |
| $R_g$ | 34.05 Å | 33.68 Å |
| $\chi^2$ | 12.53 | 1.83 |
| $c_1$ | 1.02 | 1.04 |
| $c_2$ | 1.91 | −1.77 |
| MultiFoXS | | |
| # of conformations | 10,000 | 10,000 |
| flexible residues | 82–96 | 82–96 |
| # of states | 2 | 1 |
| $R_g$ | 34.91 Å (71%) 45.69 Å (29%) | 33.28 Å |
| $\chi^2$ | 1.2 | 1.96 |
| $c_1$ | 1.0 | 0.99 |
| $c_2$ | 3.91 | −0.5 |
| **Additional EFA components** | **c1: truncated dimer** | **c0: ambiguous** |
| Guinier Analysis | | |
| $I(0)$ (Arb.) | 1.042±0.004 | 0.478±0.003 |
| $R_g$ | 28.14±0.17 Å | 42.18±0.48 Å |
| $q$-range | 0.0027–0.0461 Å$^{-1}$ | 0.005–0.0305 Å$^{-1}$ |
| $q_{max}R_g$ | 1.298 | 1.295 |
| Coefficient of correlation, $r^2$ | 0.931 | 0.948 |

*Table 1 continued on next page*

*Table 1 continued*

| | | |
|---|---|---|
| Volume (adjusted $V_P$ as SAXS MoW2) | 64,500 Å³ | 197000 Å³ |
| MW, MoW2 method (ratio to expected) | 53.6 kDa | 163.5 kDa (1.55) |
| MW, Vc method (kDa) (ratio to expected) | 48.8 kDa | 145.3 kDa (1.37) |
| *P(r)* analysis | | |
| *I*(0) (Arb.) | 1.055±0.004 | 0.486±0.004 |
| $R_g$ | 29.29±0.13 Å | 45.18±0.48 Å |
| $D_{max}$ | 98.0 Å | 124 Å |
| *q*-range | 0.0027–0.3327 Å⁻¹ | 0.0027–0.3327 Å⁻¹ |
| $\chi^2$ (total estimate from GNOM) | 1.421 (0.924) | 1.628 (0.84) |
| **Additional MultiFoXS runs (*Figure 4—figure supplement 3*)** | **S6A, N-terminus constraint** | **S6B, N-terminus constraint** |
| MultiFoXS | | |
| # of conformations | 10,000 | 10,000 |
| flexible residues | 82–86 | 82–92 |
| # of states | 2 | 2 |
| $R_g$ | 34.09 Å (62%)<br>45.20 Å (38%) | 33.54 Å (55%)<br>43.17 Å (45%) |
| $\chi^2$ | 1.26 | 1.25 |
| $c_1$ | 1.0 | 1.0 |
| $c_2$ | 4.0 | 4.0 |
| | **S6C, N-terminus constraint** | **S6D, N-terminus constraint** |
| MultiFoXS | | |
| # of conformations | 10,000 | 10,000 |
| flexible residues | 92–96 | 82–96 |
| # of states | 2 | 2 |
| $R_g$ | 41.40 Å (61%)<br>31.92 Å (39%) | 37.09 Å (47%)<br>34.08 Å (53%) |
| $\chi^2$ | 1.26 | 5.4 |
| $c_1$ | 1.01 | 1.01 |
| $c_2$ | 3.91 | 0.94 |

from $V_c$ method), which eluted earlier than a smaller component (MW 48.8 kDa calculated from $V_c$ method; *Table 1* Guinier fits, and *Figure 4—figure supplement 2*). We conclude that the larger component is a RsbU dimer, while the smaller fragment is likely a heterodimeric complex of RsbU and the C-terminal truncation product of RsbU (*Figure 4—figure supplement 1*).

Indicative of significant deviation between the RsbU structure in solution to the AlphaFold2 model, the scattering intensity profile (I(q) vs. q) was a poor fit ($\chi^2$ 12.53) to a profile calculated from the AlphaFold2 model of an RsbU dimer using FoXS (*Schneidman-Duhovny et al., 2016*; *Schneidman-Duhovny et al., 2013*; *Figure 4A*). We therefore assessed the SAXS data for the RsbU dimer for features that report on flexibility (*Kikhney and Svergun, 2015*). First, the scattering intensity data lacked distinct features caused by the multi-domain structure of RsbU from the AlphaFold2 model (*Figure 4A*). This can be visualized more readily from a dimensionless Kratky plot, which is bell-shaped and peaks above 1.3, indicating that the RsbU dimer is non-globular (*Figure 4B*). Additionally, the data do not show the two peaks from the FoXS computed model of the AlphaFold2 dimer that reflect

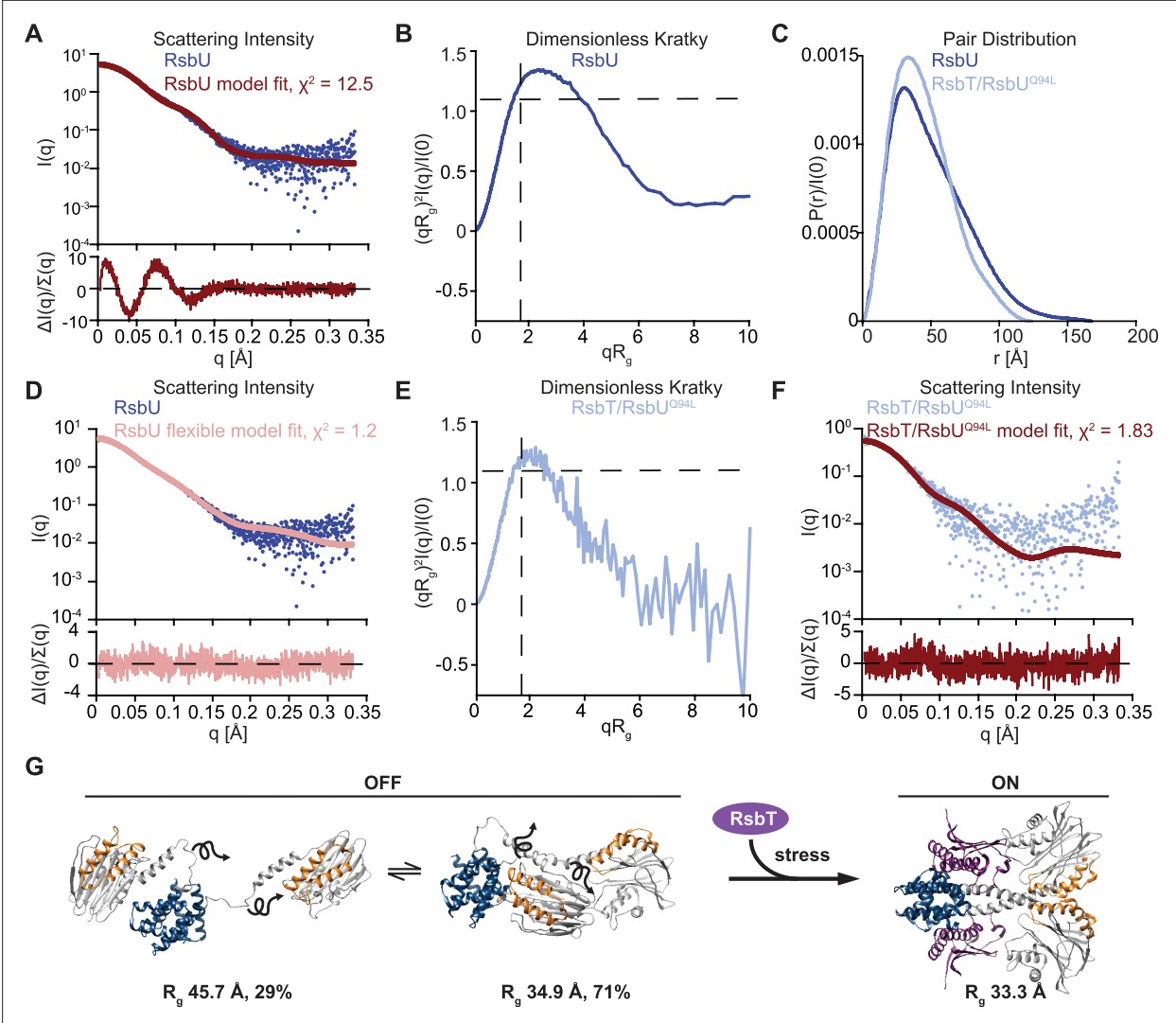

**Figure 4.** RsbU linker is flexible in the absence of RsbT, and is rigidified by RsbT binding. (**A**) I(q) versus q plot of the dimer component of SAXS scattering data from RsbU (blue) overlaid with a FoXS generated profile fit of the AlphaFold2 RsbU dimer structure (red). The lower plot shows the uncertainty normalized residuals of the fit ((experimental–computed)/error), $\chi^2$ of 12.53. (**B**) Dimensionless Kratky plot of SAXS scattering data from the dimer component of RsbU experimental data (blue). The experimental profile was logarithmically binned to reduce noise at high $qR_g$. (**C**) Normalized P(r) plot from the RsbU dimer (blue) and heterotetrameric RsbT/RsbU$^{Q94L}$ complex (light blue) is shown. (**D**) I(q) versus q plot of the dimer component of SAXS scattering data from RsbU (blue) is shown overlaid with a MultiFoXS generated two-state profile fit with flexibility allowed for residues 82–96 (pink). The lower plot shows the uncertainty normalized residuals of the fit ((experimental–computed)/error), $\chi^2$ of 1.2. (**E**) Dimensionless Kratky plot of the heterotetrameric component of RsbT/RsbU$^{Q94L}$ complex is shown binned logarithmically to reduce noise at high $qR_g$ values. (**F**) I(q) versus q plot of the heterotetrameric RsbT/RsbU$^{Q94L}$ complex (blue) overlaid with a FoXS generated profile fit of the AlphaFold2 RsbT/RsbU heterotetramer structure (red). The lower plot shows the uncertainty normalized residuals of the fits ((experimental–computed)/error), $\chi^2$ of 1.83. (**G**) Schematic of the conformational change of RsbU upon RsbT binding. The two best fit flexible models of RsbU generated from MultiFoXS in panel (**B**) are shown with the calculated $R_g$ and percentage of each model used in the fit. Squiggly arrows are shown to indicate that these models are approximate and likely represent an average of many flexible conformations.

The online version of this article includes the following source data and figure supplement(s) for figure 4:

**Figure supplement 1.** MALS estimate of molecular weight of RsbU dimer, RsbT/RsbU$^{Q94L}$ complex, RsbT/RsbU complex, and RsbU$^{Q94L}$.

**Figure supplement 1—source data 1.** Original uncropped images of the gels shown in *Figure 4—figure supplement 1B and D*.

**Figure supplement 1—source data 2.** Original image files for the gels shown in *Figure 4—figure supplement 1B and D*.

**Figure supplement 2.** EFA deconvolution and Guinier analysis of RsbU dimer SAXS profiles.

**Figure supplement 3.** Varying constraints and flexibility of RsbU model.

**Figure supplement 4.** EFA deconvolution and Guinier analysis of RsbT/RsbU$^{Q94L}$ SAXS profiles.

the extended two domain structure (*Kikhney and Svergun, 2015*; *Receveur-Brechot and Durand, 2012*; *Figure 4—figure supplement 2*). The scattering data had several additional characteristic features indicating flexibility: the P(r) plot is smooth and has an extended tail (*Cordeiro et al., 2017*; *Kikhney and Svergun, 2015*; *Receveur-Brechot and Durand, 2012*; *Figure 4C*), $R_g$ and I(0) calculated from P(r) are larger than when calculated from Guinier fits (*Cordeiro et al., 2017*; *Kikhney and Svergun, 2015*; *Receveur-Brechot and Durand, 2012*; *Table 1*), molecular weight calculated using the MoW2 modified Porod volume method is an overestimate (*Rambo and Tainer, 2011*; *Table 1*), and the $R_g$ (39.05±0.16 Å) was larger than the $R_g$ of the AlphaFold2 dimer model (34.05 Å). Prompted by this evidence of flexibility, we performed multi-state modelling using MultiFoXS (*Schneidman-Duhovny et al., 2016*), which incorporates flexibility and heterogeneity. MultiFoXS produced a two-state solution with $\chi^2$ of 1.2 (as compared to the one-state model with $\chi^2$ of 12.5) and low residuals across the entire range of data (*Figure 4D*, *Figure 4—figure supplement 2*). We allowed flexibility for the linker residues 82–96 while keeping the N-terminal domain dimerized because a stable dimer of the N-terminal domain is observed in solution, was present in a previous X-ray crystal structure of C-terminally truncated RsbU{Delumeau:2004ep}, and C-terminally truncated RsbU coelutes as a heterodimer with the full-length protein (*Figure 4—figure supplement 1*). Importantly, the precise region of allowed linker flexibility does not have a substantial impact on the MultiFoXS fit (*Figure 4—figure supplement 3*), but constraining the C-terminal phosphatase domain dimer does not yield a good fit to the data ($\chi^2$ of 5.4; *Figure 4—figure supplement 3*). The structural models predicted by MultiFoXS depict a dimerized N-terminal domain with flexible linkers and PPM domains that do not make contact. Notably, this approach also would include models in which both domains remain dimerized, suggesting that such models are not the best fit of the data. The two conformations that were selected as the best fit to the scattering data by MultiFoXS also have very different $R_g$ values, (34.91 Å [71%] and 45.69 Å [29%]), indicating RsbU samples a range of conformational space that includes a more compact state(s) consistent with the AlphaFold2 model and a significantly extended state(s), emphasizing that the linker is highly flexible. Together, we conclude that the RsbU linker appears to be flexible in the absence of RsbT (*Figure 4G*).

## RsbT rigidifies and compacts the RsbU dimer

To test the hypothesis that RsbT rigidifies the linker to produce an active RsbU dimer, we attempted SEC-MALS-SAXS analysis of the RsbT/U complex. RsbT/U eluted with molecular weight (determined by MALS) of 88 kDa compared to 105 kDa for a 2RsbT/2RsbU heterotetramer (*Figure 4—figure supplement 1*), suggesting that the RsbT/U complex had largely disassembled on the column. We therefore performed SEC-MALS-SAXS on the RsbT/RsbU$^{Q94L}$ complex, which has a higher affinity (*Figure 2A*, *Figure 4—figure supplement 4*). The RsbT/U$^{Q94L}$ complex eluted as a major peak with a higher molecular weight shoulder (*Figure 4—figure supplements 1 and 4*; also visible when we ran RsbU$^{Q94L}$ alone. *Figure 4—figure supplement 1*), with the average molecular weight of the main peak (determined by MALS) of 125 kDa (*Figure 4—figure supplement 1*). This is larger than the predicted molecular weight of a heterotetrameric 2RsbT/2RsbU$^{Q94L}$ complex (105 kDa), suggesting that some higher order complexes form. EFA deconvolution (*Meisburger et al., 2016*) identifies two components. The first deconvolved component had a SAXS determined molecular weight of 164 kDa, while the second had a SAXS molecular weight of 103 kDa, consistent with a 2RsbT/2RsbU complex (105 kDa). Together with the fact that RsbT co-elutes with RsbU$^{Q94L}$ throughout the entire peak (*Figure 4—figure supplement 1*) and the MALS molecular weight, we assess that this second component is an 2RsbT/2RsbU hetero-tetramer. The identity of the higher molecular weight component of the scattering data is ambiguous, but could represent a larger assembly of RsbT/RsbU$^{Q94L}$ or multimers of RsbU$^{Q94L}$ that were observed when RsbU$^{Q94L}$ was analyzed alone (*Figure 4—figure supplement 1*). Several lines of evidence demonstrate that the 2RsbT/2RsbU$^{Q94L}$ complex lacks the flexibility of the 2RsbU complex. First, the $R_g$ from the Guiner fit (35.06±0.43 Å) matches well with the model computed from the AlphaFold2 prediction of the heterotetrameric complex of 2RsbT/2RsbU using FoXS (33.68 Å) and is smaller than the $R_g$ we observed for the RsbU dimer (39.05±0.16 Å). Second, the features of the scattering data that suggested flexibility for RsbU dimers all suggest that the RsbT/RsbU complex is more rigid (the quality of the FoXS fit, a lower peak of the dimensionless Kratky plot, and a shorter tail of the P(r) plot; *Cordeiro et al., 2017*; *Kikhney and Svergun, 2015*; *Receveur-Brechot and Durand, 2012*; *Figure 4C, E, F*, *Figure 4—figure supplement 4*).

Third, fitting the scattering profile (I(q) vs. q) to the AlphaFold2 model yielded a $\chi^2$ of 1.83 without requirement for flexibility (*Figure 4F*). As expected, based on the lack of evidence of flexibility in the scattering profile and P(r) function, the model is not improved by the use of multi-state modeling by MultiFoXS, which was run as a control (*Schneidman-Duhovny et al., 2016*; *Figure 4—figure supplement 4*). Together, these data support the validity of the AlphaFold2 model of the heterotetrameric 2RsbT/2RsbU complex, and additionally suggest that RsbT rigidifies the otherwise flexible linker of RsbU to dimerize the phosphatase domains (*Figure 4G*). Together with our biochemical data linking RsbU linker conformation to RsbT binding, metal cofactor interaction, and phosphatase activity, this provides a comprehensive model for how RsbU is activated (*Figure 4G*).

## Regulated dimerization is an evolutionarily adaptable mechanism to control bacterial phosphatases

*B. subtilis* has a second GSR-initiating PPM phosphatase, RsbP, that has an N-terminal PAS domain and dephosphorylates RsbV to activate σ$^B$ in response to energy stress. Our AlphaFold2 prediction of an RsbP dimer shows an extended alpha-helical linker connecting dimeric PAS domains and phosphatase domains. Like RsbU, the interface between the dimerized phosphatase domains is predicted to be mediated by α1 and α2 of the phosphatase domain. We found that previously identified constitutively activating mutations in RsbP (*Brody et al., 2009*) map to similar positions as the gain-of-function mutations we identified here for RsbU in their respective predicted structures, namely the interface between α1 and α2 of the phosphatase domains and the linker. RsbP is activated by a partner-protein, RsbQ, that has analogous function to RsbT for RsbU. An AlphaFold2 prediction of the heterotetrameric 2RsbP/2RsbQ complex places the two molecules of RsbQ in a hybrid interface between the PAS domains and linkers, similar to how RsbT contacts RsbU. These predictions suggest a testable hypothesis that RsbP is controlled through an activation mechanism similar to that of RsbU (*Figure 5A*).

To further probe the generality of the linker-mediated phosphatase dimer we observe for RsbU, we generated AlphaFold2 predictions for dimers of other documented GSR initiating phosphatases (*C. difficile* RsbZ [N-terminal tandem GAF domains; *Figure 5B*], *S. coelicolor* OsaC [N-terminal GAF and PAS domains], and *M. tuberculosis* Rv1364c [N-terminal PAS domain, C-terminally fused to an anti-σ factor domain and an anti-anti-σ factor domain]), as well as from examples of common domain architectures of PPM phosphatases selected from InterPro (response receiver [*Streptomyces coelicolor*, MY40396.1], GAF [*Rhodococcus aetherivorans*, WP_029544106.1], and HAMP [*Synechocystis sp.*, WP_010874143.1] domains; *Figure 5—figure supplement 1*). In each case, AlphaFold2 produced high-confidence predictions of phosphatase domains dimerized through a shared α0, α1, α2 interface, connected to dimeric N-terminal domains by crossing α-helical linkers (*Figure 1—figure supplements 1–2*, *Figure 5—figure supplement 1*). Consistent with a model in which the stability of the linker plays a conserved regulatory role, the AlphaFold2 models for many of the predicted structures have unfavorable polar residues buried in the coiled-coil interface (positions *a* and *d*, for which non-polar residues are most favorable; *Figure 5—figure supplement 2*). From this analysis, we speculate that linker-mediated phosphatase domain dimerization is an evolutionarily conserved, adaptable mechanism to control PPM phosphatase activity. This mechanistic/structural congruence provides a framework for how distantly related phosphatases can be regulated to control diverse biological processes in bacteria (*Figure 5E*).

## Discussion

Since the ground-breaking discovery of σ$^B$, the first conditionally activated cellular transcription factor to be identified, 45 years ago (*Haldenwang and Losick, 1979*), a persistent unanswered question has been how σ$^B$ is turned on. Here, our findings reveal how RsbU is activated as a phosphatase in the decisive step of σ$^B$ activation. Below, we describe how this mechanism is broadly applicable to other bacterial signaling pathways.

### RsbT activates RsbU through a conserved phosphatase dimer

Using unbiased genetic screens, biochemical reconstitution, biophysical approaches, and structural prediction, we have discovered that RsbT activates RsbU by rigidifying an otherwise flexible linker region to allosterically activate RsbU as a phosphatase. Our structural prediction, supported by

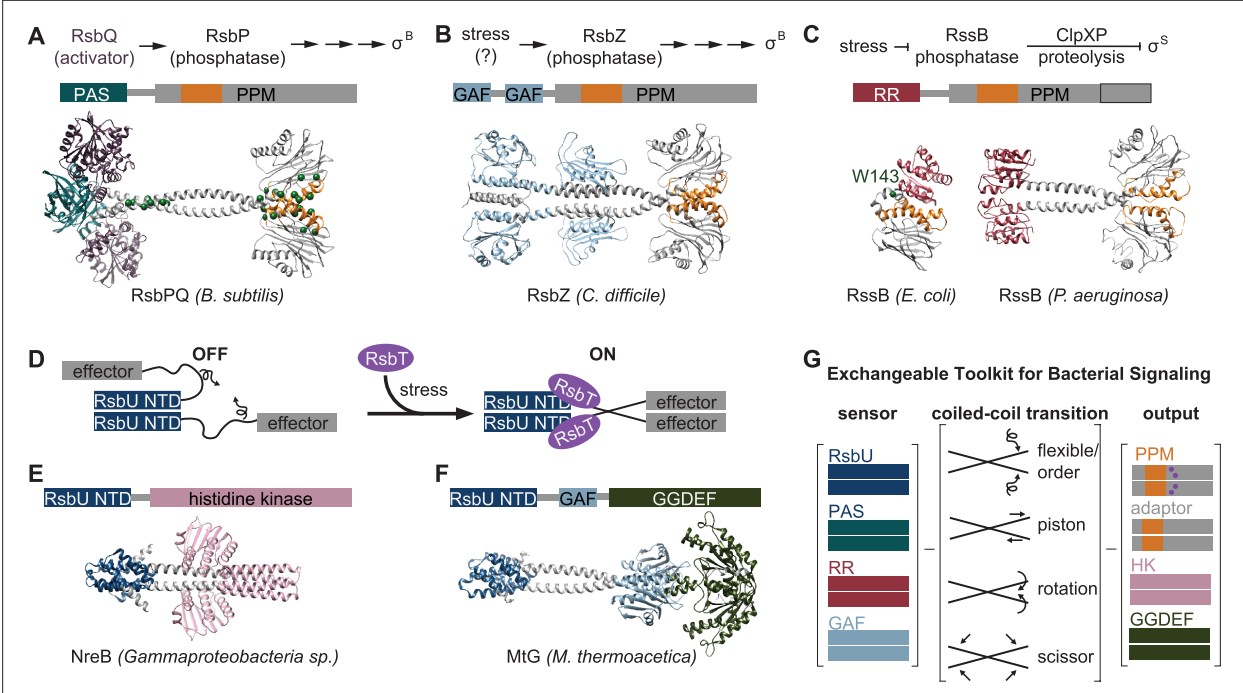

**Figure 5.** An exchangeable toolkit for bacterial signal transduction. (**A**) An AlphaFold2 model of the *B. subtilis* energy stress phosphatase RsbP is shown bound to its activating α/β-hydrolase protein RsbQ (purple) with a schematic of the σB activation pathway shown above. RsbP has an N-terminal PAS domain (turquoise) that is connected to the PPM phosphatase domain by a predicted coiled coil linker. The Cα positions of bypass suppressor mutations that render RsbP active in the absence of RsbQ (***Brody et al., 2009***) are shown as green spheres. The switch element of the PPM phosphatase domain (α1 and α2) is colored orange. (**B**) An AlphaFold2 model of the *C. difficile* GSR phosphatase RsbZ is shown with a schematic of the σB activation pathway shown above. RsbZ has tandem N-terminal GAF domains (blue) that connect to the PPM phosphatase domain by a predicted coiled coil linker. The switch element of the PPM phosphatase domain (α1 and α2) is colored orange. (**C**) Experimental structures of the GSR initiating adaptor protein RssB from *E. coli* (left, 8T85) and *P. aeruginosa* (right, 3F7A) are shown with a schematic of the σS activation pathway shown above. The Cα positions of a bypass suppressor mutation (W143R) that renders RssB insensitive to anti-adapter proteins is shown as a green sphere. The switch element of the PPM phosphatase domain (α1 and α2) is colored orange. The black outline shows the portion of the PPM phosphatase domain that is missing in the *E. coli* RssB adaptor protein, which lacks phosphatase activity. (**D**) A cartoon schematic of activation by RsbT of proteins with RsbU NTD through ordering of a flexible linker. The protein is depicted with dimerized RsbU N-terminal dimerization domains (blue) with flexible linkers (black) that extend into two effector domains (grey). During stress conditions, RsbT (purple) activates the protein through binding to the N-terminal dimerization domains and linkers, rigidifying the linkers to dimerize the effector domains. (**E**) An AlphaFold2 model of a dimer of the histidine kinase NreB from a *Gammaproteobacteria* species (MBI3545564.1) is shown with the RsbU N-terminal domain colored blue and the histidine kinase domain colored pink. (**F**) An AlphaFold2 model of a dimer of the GGDEF diguanylate cyclase MtG from *M. thermoacetica* (WP_011392981.1) is shown with RsbU N-terminal domain colored blue, GAF domain colored lite blue, and GGDEF domain colored green. (**G**) Schematic illustrating how various sensor domains (RsbU N-terminal dimerization domain, PAS, response receiver (RR), and GAF) can control various output activities serine/threonine phosphatase (PPM), histidine kinase (HK), protease adaptor, and diguanylate cyclase (GGDEF). We hypothesize that the shared regulatory mechanism through a dimeric coiled-coil linker makes these regulatory domains modularly exchangeable across effector domains. Known mechanisms for how allosteric regulation is transmitted through the linker are listed and shown with arrow diagrams.

The online version of this article includes the following figure supplement(s) for figure 5:

**Figure supplement 1.** AlphaFold2 predictions of varying signaling PPM phosphatases.

**Figure supplement 2.** Coiled-coil analysis of structures of bacterial signaling enzymes.

our genetic and structural data suggest that RsbU linkers assume a crossed α-helical conformation that dimerizes the phosphatase domains (***Figure 4G***). This dimerization activates RsbU through a conserved switch element at the dimer interface, decreasing the $K_M$ for metal cofactor and increasing the catalytic rate. Our structural predictions suggest that this is a widespread mechanism to regulate the activity of bacterial protein serine/threonine phosphatases. While RsbU activation is controlled by a flexible-to-rigid transformation of the linker, it is possible that other conformational changes of the linker could be alter the PPM dimer to control phosphatase activity through the conserved switch element.

## A general mechanism for general stress response activation

*Escherichia coli* and *B. subtilis* are the most extensively studied models for GSR activation (*Battesti et al., 2011*; *Hecker et al., 2007*). However, key mechanistic features of the activation mechanisms differ between the two systems. The *E. coli* GSR is initiated by stabilizing the alternative sigma factor $\sigma^S$ (RpoS) against degradation by ClpXP (*Figure 5C*). Under non-stressed conditions, degradation of $\sigma^S$ is mediated by the adapter protein, RssB, which is inhibited by anti-adaptors (IraD, IraM, and IraP) and stimulated by phosphorylation on its N-terminal response-receiver domain during stress. Historically, the differences between these systems were interpreted to mean that they operate by distinct mechanistic principles. However, RssB contains a C-terminally truncated (and therefore inactive) PPM phosphatase domain, suggesting some aspects of its regulatory mechanism could be shared with RsbU (*Battesti et al., 2013*).

Our findings reveal that *E. coli* RssB and *B. subtilis* RsbU use three shared regulatory features to control GSR activation through their distinct downstream mechanisms. First, the positioning of analogous flexible linkers controls GSR activation by RssB and RsbU (*Brugger et al., 2023a*; *Brugger et al., 2023b*; *Dorich et al., 2019*). The C-terminal pseudo-phosphatase domain of RssB is joined to its N-terminal $\sigma^S$ binding response-receiver domain by a predominantly α-helical linker (termed the segmented helical linker, or SHL) of similar length to the linker of RsbU (*Dorich et al., 2019*). The SHL folds to occlude the $\sigma^S$ interacting face of the response-receiver domain when bound to anti-adapters, and is remodeled to promote $\sigma^S$ interaction when the response-receiver domain is phosphorylated (*Brugger et al., 2023a*; *Brugger et al., 2023b*; *Dorich et al., 2019*). Supporting the hypothesis that the SHL is mechanistically related to the RsbU linker, substitution of W143, the homologous residue to Q94 of *B. subtilis* RsbU, blocks regulation by IraD and IraP (*Battesti et al., 2013*). Second, the interface between the linker and α1 of the phosphatase domain is critical for regulation of both RssB and RsbU (*Brugger et al., 2023a*; *Dorich et al., 2019*). W143 packs in a hydrophobic pocket formed with the α1 helix of the pseudo-phosphatase domain of phosphorylated RssB and IraD-bound RssB (*Brugger et al., 2023a*; *Dorich et al., 2019*). Substitution of residues in α1, including A216 (corresponding to M166 of RsbU), block regulation of RssB by anti-adapters, underscoring the mechanistic similarity (*Battesti et al., 2013*). Third, conformational change of the switch helices (α1 and α2) of the phosphatase domain is of shared regulatory importance (*Battesti et al., 2013*). HDX experiments demonstrate that there is decreased exchange of the α1 helix of the pseudo-phosphatase domain of RssB when $\sigma^S$ is bound, suggesting that the conformational dynamics of this region are reduced by substrate recognition (*Brugger et al., 2023a*).

These striking mechanistic similarities between RssB from *E. coli* and RsbU from *B. subtilis* suggest that key regulatory features were conserved when an ancestral phosphatase lost enzymatic activity and became a dedicated adapter protein in *E. coli* and related γ-proteobacteria. Interestingly, *P. aeruginosa* has a homologue of RssB/RsbU that functions as an adapter for $\sigma^S$ degradation, yet has a catalytically active, full-length phosphatase domain and resides in an operon with an *rsbV* homologue (*rssC*; *Rodríguez-Martínez et al., 2023*). Crystal structures of *P. aeruginosa* RssB revealed that it forms a dimer with crossing linkers and dimerized phosphatase domains with striking similarity to the dimer predicted for RsbU (*Figure 5*, *Figure 5—figure supplement 2C*). Intriguingly, adapter protein function of *P. aeruginosa* RssB requires RssC (*Rodríguez-Martínez et al., 2023*), and *E. coli* RssB has been observed to weakly dimerize (*Dorich et al., 2019*), further supporting a direct mechanistic link between phosphatase regulation and adapter protein function. While not all GSR regulatory mechanisms involve a phosphatase or adapter protein related to RsbU/RssB, these findings suggest that linker-mediated control of a PPM phosphatase is a broadly conserved mechanism for GSR initiation across bacterial phyla.

## An exchangeable toolkit for signal transduction

Histidine kinases and GGDEF diguanylate cyclases are the two most prevalent families of bacterial signal transduction proteins. A striking feature of their regulatory mechanisms is that they are often active in dimeric complexes that are allosterically controlled by N-terminal regulatory domains that propagate signals through crossing α-helical linkers (*Zschiedrich et al., 2016*; *Randall et al., 2022*; *Gourinchas et al., 2017*). Our findings that the crossing α-helical linkers are structurally similar (Fig. S2C) and that shared regulatory domains (response receiver, HAMP, and PAS domains) connect phosphatase dimers by a coiled-coil linker (*Figure 5A-C*, *Figure 5—figure supplement 1*) suggests that

this shared allosteric regulatory architecture allows modular exchange of N-terminal domains between PPM phosphatases, histidine kinases, and GGDEF diguanylate cyclases (*Figure 5D*). In support of this, the RsbU N-terminal domain and RsbT were previously shown to control GGDEF diguanylate cyclase activity in *Moorella thermoacetica* (*Quin et al., 2012*), histidine kinases with RsbU N-terminal domains are found in operons with *rsbT* and stressosome genes in some *Proteobacteria* species (*Heinz et al., 2022*), and AlphaFold2 prediction of these dimeric complexes reveals coiled-coil linkers to the effector domains (*Figure 5E–F*). Similarly, the PAS/α/β hydrolase module that controls RsbP phosphatase activity in *B. subtilis* is found coupled to histidine kinases and GGDEF diguanylate cyclases across bacterial phyla (*Nadezhdin et al., 2011*).

While the specific mechanism discovered here for linker control of RsbU activity (flexibility controlled by a binding partner) is distinct from the models for linker control of histidine kinases and GGDEF diguanylate cyclases (piston shift, scissoring, and helical rotation; *Zschiedrich et al., 2016*; *Randall et al., 2022*), the shared dimeric architecture suggests that N-terminal regulatory domains could be modularly exchanged across effector domain families (*Figure 5G*). Allosteric control of PPM switch element conformation could theoretically be achieved by piston shift, scissoring, or helical rotation. Future characterization of PPM phosphatases will be required to determine how widespread linker flexibility is as a regulatory mechanism and how modularly sensory domains can be exchanged across signaling effector families.

# Materials and methods
## Protein expression and purification
All proteins were expressed in *E. coli* BL21 (DE3) cells grown at 37 °C to an $OD_{600}$ of 0.4 and induced at 16 °C for 14–18 hr with 1 mM isopropyl β-d-1-thiogalactopyranoside (IPTG) unless otherwise specified. Cells were harvested and purified as follows:

### RsbU and RsbU variants
Cell pellets were resuspended in lysis buffer with 1 mM phenylmethylsulfonyl fluoride (PMSF; 50 mM K•HEPES, pH 7.5, 100 mM NaCl, 20 mM imidazole, 10% glycerol, 0.5 mM dithiothreitol (DTT)), and were lysed using two passes in a microfluidizer at 10,000 PSI. Cell lysates were cleared by spinning at 16,000 RPM for 45 min in an Avanti JA-20 rotor. Cleared lysates were then run over a HisTrap HP column on an AKTA FPLC, washed with lysis buffer, and eluted with a gradient to 200 mM imidazole. RsbU was run on a Superdex200 16/600 column equilibrated with lysis buffer on an AKTA FPLC. Fractions were pooled and the 6-His tags were cleaved with 3 C protease in dialysis to lysis buffer overnight at 4 °C. Cleaved tags and 3 C protease were subtracted by passing over a column containing equilibrated Ni-NTA resin. 2 mM EDTA was added to cleaved protein prior to gel filtration run. RsbU was then further purified on a Superdex200 16/600 column equilibrated with 20 mM K•HEPES pH 7.5, 100 mM NaCl, 2 mM DTT on an AKTA FPLC. Fractions were pooled, concentrated to 200 µM, and flash-frozen and stored at –80 °C.

### RsbT/U and RsbT/RsbU$^{Q94L}$ co-expression
Cell pellets were resuspended in lysis buffer with 1 mM PMSF (50 mM K•HEPES pH 8.0, 200 mM NaCl, 20 mM imidazole, 10% glycerol, 0.5 mM DTT), and were lysed using two passes in a microfluidizer at 10,000 PSI. Cell lysates were cleared by spinning at 16,000 RPM for 45 min in an Avanti JA-20 rotor. Cleared lysates were then run over a HisTrap HP column on an AKTA FPLC, washed with lysis buffer, and eluted with a gradient to 200 mM imidazole. 2 mM EDTA was added to protein prior to gel filtration run. RsbT/RsbU was then further purified on a Superdex200 16/600 column equilibrated with 20 mM K•HEPES pH 8.0, 100 mM NaCl, 2 mM DTT on an AKTA FPLC. Fractions were pooled, concentrated to 200 µM, and flash-frozen and stored at –80 °C.

### RsbT
Cells were resuspended in lysis buffer with 1 mM PMSF (50 mM K•HEPES pH 8.0, 100 mM NaCl, 20 mM imidazole, 10% glycerol, 2 mM $MgCl_2$, 0.1 mM ATP, 0.5 mM DTT), and were lysed using two passes in a microfluidizer at 10,000 PSI. Cell lysates were cleared by spinning at 16,000 RPM for 45 min in an Avanti JA-20 rotor. Cleared lysates were then run over 2 mL/L of cells of Ni-NTA resin

slurry equilibrated with lysis buffer. The resin was then washed with 10 CV of lysis buffer. Ni-NTA resin was resuspended with 10 mL of lysis buffer with 400 mM imidazole, and was incubated for 10 min. RsbT was eluted off column, elution was tracked by Bradford Reagent. Additional elution buffer was added until all protein was eluted off column. The 6-His tags were cleaved with 3 C protease in dialysis to lysis buffer overnight at 4 °C. Cleaved tags and 3 C protease were subtracted by passing over a column containing equilibrated Ni-NTA resin. Protein was concentrated to approximately 50 µM, was flash-frozen and stored at –80 °C.

### RsbV

RsbV was expressed in *E. coli* BL21 (DE3) cells grown at 37 °C to an $OD_{600}$ of 0.4 and induced at 37 °C for 3–4 hrs with 1 mM IPTG. Cells were harvested and resuspended in lysis buffer with 1 mM PMSF (50 mM K•HEPES pH 8.0, 200 mM NaCl, 20 mM imidazole, 10% glycerol, 0.5 mM DTT), and were lysed using two passes on a microfluidizer at 10,000 PSI. Cell lysates were cleared by spinning at 16,000 RPM for 45 min in an Avanti JA-20 rotor. Cleared lysates were then run over a HisTrap HP column on an AKTA FPLC, washed with lysis buffer, and eluted with a gradient to 200 mM imidazole. 2 mM EDTA was added to protein prior to gel filtration run. The 6-His tags were cleaved with 3 C protease in dialysis to lysis buffer overnight at 4 °C. Cleaved tags and 3 C protease were subtracted by passing over a column containing equilibrated Ni-NTA resin. Cleaved protein was then further purified on a Superdex75 16/60 equilibrated in 20 mM K•HEPES pH 7.5, 150 mM NaCl, 10% glycerol, 2 mM DTT. Fractions were pooled, concentrated to approximately 100 µM, and flash-frozen and stored at –80 °C.

### RsbW

Cell pellets were resuspended in lysis buffer with 1 mM PMSF (20 mM K•HEPES pH 7.5, 200 mM NaCl, 20 mM imidazole, 10 mM $MgCl_2$, 10% glycerol, and 0.5 mM DTT), and were lysed using two passes on a microfluidizer at 10,000 PSI. Cell lysates were cleared by spinning at 16,000 RPM for 45 min in an Avanti JA-20 rotor. Cleared lysates were then run over a HisTrap HP column on an AKTA FPLC, washed with lysis buffer, and eluted with a gradient to 200 mM imidazole. The 6 H tag was left uncleaved to aid removal after phosphorylation reactions. RsbW was then further purified on a Superdex75 16/60 equilibrated in 50 mM K•HEPES pH 7.5, 150 mM NaCl, 10% glycerol, and 2 mM DTT. Fractions were pooled, concentrated to approximately 100 µM, and flash-frozen and stored at –80 °C.

### RsbV-P

RsbVW coexpression plasmid was expressed in *E. coli* BL21 (DE3) cells grown at 37 °C to an $OD_{600}$ of 0.4 and induced at 37 °C for 3–4 hr with 1 mM IPTG. Cells were harvested and resuspended in lysis buffer with 1 mM PMSF (50 mM K•HEPES pH 7.5, 50 mM KCl, 20 mM imidazole, 0.5 mM DTT), and were lysed using two passes on a microfluidizer at 10,000 PSI. Cell lysates were cleared by spinning at 16,000 RPM for 45 min in an Avanti JA-20 rotor. Cleared lysates were then run over a HisTrap HP column on an AKTA FPLC, washed with lysis buffer, and eluted with a gradient to 200 mM imidazole. The 6-His tags were cleaved with 3 C protease in dialysis to lysis buffer overnight at 4 °C. Cleaved tags and 3 C protease were subtracted by passing over a column containing equilibrated Ni-NTA resin. RsbVW was run on a Superdex75 16/60 column equilibrated with lysis buffer on an AKTA FPLC. Fractions of RsbVW coelution were pooled and RsbV was phosphorylated in dialysis at room temperature in lysis buffer with 5 mM ATP and 10 mM $MgCl_2$. RsbV-P Fractions were pooled, concentrated to 200 µM, and flash-frozen and stored at –80 °C. Dialyzed RsbVW were then concentrated and run on Superdex75 16/60 column equilibrated with 20 mM K•HEPES pH 7.5, 150 mM NaCl, 10% glycerol, 2 mM DTT. RsbV-P Fractions were pooled, concentrated to 200 µM, and flash-frozen and stored at –80 °C.

## SEC-MALS-SAXS

Small Angle X-ray Scattering (SAXS) was performed at BioCAT (beamline 18ID at the Advanced Photon Source, Chicago) with in-line size exclusion chromatography (SEC) to separate sample from aggregates and other contaminants and ensure optimal sample quality. In-line multiangle light scattering (MALS), dynamic light scattering (DLS) and differential refractive index (dRI) measurements were recorded for additional biophysical characterization (SEC-MALS-SAXS). The samples were loaded on a Superdex

200 10/300 Increase GL Column (Cytiva) run by a 1260 Infinity II HPLC (Agilent Technologies) at 0.6 mL/min. The flow passed in order through an Agilent UV detector, a MALS detector and a DLS detector (DAWN Helios II, Wyatt Technologies), and an RI detector (Optilab T-rEX, Wyatt) before the SAXS flow cell. The flow cell consists of a 1.0 mm ID quartz capillary with ~20 μm walls. A coflowing buffer sheath is used to separate sample from the capillary walls, helping prevent radiation damage (*Kirby et al., 2016*). Scattering intensity was recorded using a Pilatus3 X1 M (Dectris) detector that was placed 3.682 m from the sample giving us access to a q-range of 0.0027 Å⁻¹ to 0.33 Å⁻¹. 0.5 s exposures were acquired every 1 s during elution and data was reduced using BioXTAS RAW 2.1.4 (*Hopkins et al., 2017*; Hopkins 2024). Buffer blanks were created by averaging regions flanking the elution peak and subtracted from all measured profiles. Because of the heterogeneity in the samples, EFA was used to deconvolve the scattering profiles of the overlapping components (*Meisburger et al., 2016*). These scattering profiles were used for subsequent analysis in RAW. The quality of the deconvolution was assessed based on the mean weighted $\chi^2$ for each deconvoluted component and the $\chi^2$ of the combined fit to each frame of the dataset (*Figure 4—figure supplement 2B–E*, *Figure 4—figure supplement 4B–E*). RAW was used to run GNOM from the ATSAS package (version 3.2.1) (*Svergun, 1992*; *Manalastas-Cantos et al., 2021*). Description of each SAXS sample and results of the analysis are presented in (*Table 1*; *Trewhella et al., 2017*). The composition of each scattering component was determined based on the SAXS derived molecular weights, the MALS profile, and gels from the purification of the protein or protein complex (*Figure 4—figure supplement 1*; *Table 1*). Molecular weights and hydrodynamic radii were calculated from the MALS and DLS data respectively using the ASTRA 7 software (Wyatt). All samples were run in 20 mM HEPES pH 7.5, 100 mM NaCl, 5 mM DTT. Wild-type RsbU was run at a concentration of 3.5 mg/mL, RsbT/RsbU^Q94L was run at a concentration of 2.5 mg/mL, RsbU^Q94L was run at a concentration of 3.5 mg/mL, RsbT/RsbU was run at a concentration of 2.5 mg/mL. FoXS was used to calculated predicted scattering profiles from the model, and was compared to the experimental scattering profiles (*Schneidman-Duhovny et al., 2016*; *Schneidman-Duhovny et al., 2013*). Output parameters calculated using FoXS are provided in *Table 1*. MultiFoXS was used to generate a flexible structure ensemble with multicomponent fits and was compared to the experimental scattering profiles (*Schneidman-Duhovny et al., 2016*). MultiFoXS input and output parameters are provided in *Table 1*.

## Fluorescence anisotropy

We site-specifically labeled RsbT with tetramethylrhodamine 5-maleimide (TMR) dye at its unique cysteine (position 6). We added increasing concentrations of RsbU and RsbU^Q94L to 200 nM RsbT^TMR and measured the change in anisotropy (*Figure 2A*). All measurements were done at 25 °C in 50 mM K•HEPES pH 7.5, 100 mM NaCl, and 10% glycerol. All measurements of spectra were conducted with a slit width of 7 nm, and anisotropy measurements were conducted with a slit width of 7 nm. The excitation wavelength was 551 nm and the emission wavelength was 577 nm. Anisotropy measurements were taken 30 s apart for six measurements, and then were averaged and plot against its corresponding concentration of RsbU. Anisotropy data was then fit to a quadratic binding curve using KaleidaGraph, reported errors are the error of the fit.

## Phosphatase assays

Phosphatase assays were performed with RsbV-P that was labeled with ³²P by incubating RsbV (40 μM), 6His-RsbW (45 μM), and 100 μCi of γ-³²P ATP overnight at room temperature in 50 mM K•HEPES pH 7.5, 50 mM KCl, 10 mM MgCl₂, and 2 mM DTT. Unincorporated nucleotide was removed by buffer exchange using a Zeba spin column equilibrated in 50 mM K•HEPES pH 8.0, 100 mM NaCl. 6H-RsbW was then removed by Ni-NTA resin equilibrated in 50 mM K•HEPES pH 8.0, 100 mM NaCl, 20 mM imidazole. The flow-through fraction from the Ni-NTA resin containing RsbV-³²P was then exchanged into 50 mM K•HEPES pH 8.0, 100 mM NaCl buffer using 3 subsequent Zeba spin columns to remove all unincorporated nucleotide and free phosphate. Labeled RsbV-³²P was aliquoted and frozen at –80 °C for future use.

All phosphatase assays were performed at room temperature in 50 mM K•HEPES pH 7.5, 100 mM NaCl. The concentrations of enzyme, substrate, MnCl₂, and MgCl₂ were varied as indicated. 10 μM RsbT was additionally added to reactions as indicated. Reactions were stopped with 0.5 M EDTA, pH 8.0 and run on PEI-Cellulose TLC plates developed in 1 M LiCl₂ and 0.8 M acetic acid and imaged

**Table 2.** *B. subtilis* strains.

All *B. subtilis* strains are in the background of PY79.

| Strain # | Genotype | Reference |
|----------|----------|-----------|
| RB01 | *ΔrsbPQ ΔrsbU rsbV-FLAG amyE::ctc-lacZ Pspank-rsbU* | *Ho and Bradshaw, 2021* |
| RB02 | *ΔrsbPQ ΔrsbTU rsbV-FLAG amyE::ctc-lacZ Pspank-rsbU* | *Ho and Bradshaw, 2021* |
| RB03 | *ΔrsbPQ ΔrsbTU rsbV-FLAG amyE::ctc-lacZ Pspank-rsbTU* | *Ho and Bradshaw, 2021* |
| RB04 | *ΔrsbPQ ΔrsbU rsbV-FLAG amyE::ctc-lacZ Pspank-rsbU$^{Q94L}$* | This study |
| RB05 | *ΔrsbPQ ΔrsbTU rsbV-FLAG amyE::ctc-lacZ Pspank-rsbU$^{Q94L}$* | This study |
| RB06 | *ΔrsbPQ ΔrsbU rsbV-FLAG amyE::ctc-lacZ Pspank-rsbU$^{M166V}$* | *Ho and Bradshaw, 2021* |
| RB07 | *ΔrsbPQ ΔrsbTU rsbV-FLAG amyE::ctc-lacZ Pspank-rsbU$^{M166V}$* | *Ho and Bradshaw, 2021* |
| RB08 | *ΔrsbPQ ΔrsbTU rsbV-FLAG amyE::ctc-lacZ Pspank-rsbTU$^{Y28I}$* | This study |
| RB09 | *ΔrsbPQ ΔrsbTU rsbV-FLAG amyE::ctc-lacZ Pspank-rsbTU$^{Y28I,S49G}$* | This study |
| RB10 | *ΔrsbPQ ΔrsbTU rsbV-FLAG amyE::ctc-lacZ Pspank-rsbTU$^{Y28I,K53E}$* | This study |
| RB11 | *ΔrsbPQ ΔrsbTU rsbV-FLAG amyE::ctc-lacZ Pspank-rsbTU$^{Y28I,T89A}$* | This study |
| RB12 | *ΔrsbPQ ΔrsbTU rsbV-FLAG amyE::ctc-lacZ Pspank-rsbTU$^{Y28I,G92R}$* | This study |
| RB13 | *ΔrsbPQ ΔrsbTU rsbV-FLAG amyE::ctc-lacZ Pspank-rsbTU$^{Y28I,G92E}$* | This study |
| RB14 | *ΔrsbPQ ΔrsbTU rsbV-FLAG amyE::ctc-lacZ Pspank-rsbTU$^{Y28I,Q94L}$* | This study |
| RB15 | *ΔrsbPQ ΔrsbTU rsbV-FLAG amyE::ctc-lacZ Pspank-rsbTU$^{Y28I,T110S}$* | This study |
| RB16 | *ΔrsbPQ ΔrsbTU rsbV-FLAG amyE::ctc-lacZ Pspank-rsbTU$^{Y28I,V142A}$* | This study |
| RB17 | *ΔrsbPQ ΔrsbTU rsbV-FLAG amyE::ctc-lacZ Pspank-rsbTU$^{Y28I,K143E}$* | This study |
| RB18 | *ΔrsbPQ ΔrsbTU rsbV-FLAG amyE::ctc-lacZ Pspank-rsbTU$^{Y28I,C165Y}$* | This study |
| RB19 | *ΔrsbPQ ΔrsbTU rsbV-FLAG amyE::ctc-lacZ Pspank-rsbTU$^{Y28I,M173I}$* | This study |
| RB20 | *ΔrsbPQ ΔrsbTU rsbV-FLAG amyE::ctc-lacZ Pspank-rsbTU$^{R91E}$* | This study |
| RB21 | *ΔrsbPQ ΔrsbTU rsbV-FLAG amyE::ctc-lacZ Pspank-rsbT$^{D92K}$U* | This study |
| RB22 | *ΔrsbPQ ΔrsbTU rsbV-FLAG amyE::ctc-lacZ Pspank-rsbT$^{D92K}$U$^{R91E}$* | This study |
| RB23 | *ΔrsbPQ ΔrsbU rsbV-FLAG amyE::ctc-lacZ Pspank-rsbU$^{Q94I}$* | This study |
| RB24 | *ΔrsbPQ ΔrsbTU rsbV-FLAG amyE::ctc-lacZ Pspank-rsbU$^{Q94I}$* | This study |
| RB25 | *ΔrsbPQ ΔrsbU rsbV-FLAG amyE::ctc-lacZ Pspank-rsbU$^{Q94V}$* | This study |
| RB26 | *ΔrsbPQ ΔrsbTU rsbV-FLAG amyE::ctc-lacZ Pspank-rsbU$^{Q94V}$* | This study |
| RB27 | *ΔrsbPQ ΔrsbU rsbV-FLAG amyE::ctc-lacZ Pspank-rsbU$^{Q94Y}$* | This study |
| RB28 | *ΔrsbPQ ΔrsbTU rsbV-FLAG amyE::ctc-lacZ Pspank-rsbU$^{Q94Y}$* | This study |
| RB29 | *ΔrsbPQ ΔrsbU rsbV-FLAG amyE::ctc-lacZ Pspank-rsbU$^{Q94F}$* | This study |
| RB30 | *ΔrsbPQ ΔrsbTU rsbV-FLAG amyE::ctc-lacZ Pspank-rsbU$^{Q94F}$* | This study |
| RB31 | *ΔrsbPQ ΔrsbU rsbV-FLAG amyE::ctc-lacZ Pspank-rsbU$^{Q94W}$* | This study |
| RB32 | *ΔrsbPQ ΔrsbTU rsbV-FLAG amyE::ctc-lacZ Pspank-rsbU$^{Q94W}$* | This study |
| RB33 | *ΔrsbPQ ΔrsbU rsbV-FLAG amyE::ctc-lacZ Pspank-rsbU$^{Q94M}$* | This study |
| RB34 | *ΔrsbPQ ΔrsbTU rsbV-FLAG amyE::ctc-lacZ Pspank-rsbU$^{Q94M}$* | This study |
| RB35 | *ΔrsbPQ ΔrsbU rsbV-FLAG amyE::ctc-lacZ Pspank-rsbU$^{Q94N}$* | This study |
| RB36 | *ΔrsbPQ ΔrsbTU rsbV-FLAG amyE::ctc-lacZ Pspank-rsbU$^{Q94N}$* | This study |
| RB37 | *ΔrsbPQ ΔrsbU rsbV-FLAG amyE::ctc-lacZ Pspank-rsbU$^{Q94E}$* | This study |
| RB38 | *ΔrsbPQ ΔrsbTU rsbV-FLAG amyE::ctc-lacZ Pspank-rsbU$^{Q94E}$* | This study |

*Table 2 continued on next page*

*Table 2 continued*

| Strain # | Genotype | Reference |
|---|---|---|
| RB39 | ΔrsbPQ ΔrsbU rsbV-FLAG amyE::ctc-lacZ Pspank-rsbU$^{Q94A}$ | This study |
| RB40 | ΔrsbPQ ΔrsbTU rsbV-FLAG amyE::ctc-lacZ Pspank-rsbU$^{Q94A}$ | This study |
| RB41 | ΔrsbPQ ΔrsbU rsbV-FLAG amyE::ctc-lacZ Pspank-rsbU$^{Q94G}$ | This study |
| RB42 | ΔrsbPQ ΔrsbTU rsbV-FLAG amyE::ctc-lacZ Pspank-rsbU$^{Q94G}$ | This study |
| RB43 | ΔrsbPQ ΔrsbU rsbV-FLAG amyE::ctc-lacZ Pspank-rsbU$^{T89A}$ | This study |
| RB44 | ΔrsbPQ ΔrsbU rsbV-FLAG amyE::ctc-lacZ Pspank-rsbU$^{C165Y}$ | This study |

on a Typhoon scanner. Phosphatase assays were performed more than three independent times as separate experiments. Multiple turnover reactions were performed with trace RsbV-$^{32}$P and an excess of cold RsbV-P. Number of turnovers were calculated for each timepoint from (1 – fraction substrate)*[substrate]/[enzyme]. Data shown in figures is from a single representative experiment, and reported errors are the error from the fit.

## *B. subtilis* strain construction

*B. subtilis* cells were grown in Lennox lysogeny broth (LB, Sigma-Aldrich) as liquid medium or as plates supplemented with 15% Bacto Agar (Difco). Antibiotics were used as appropriate for maintenance of plasmids or for selection of transformants: MLS (50 µg/mL erythromycin, 250 µg/mL lincomycin), kanamycin (10 µg/mL). For plates used to visualize σ$^B$ reporter activity, 80 µg/mL of 5-bromo-4-chloro-3-indolyl-β-D-galactopyranoside (X-gal) and 1 mM IPTG were added. All *B. subtilis* strains were derived from the PY79 strain background and are listed in the table of strains. Plasmids based on pHB201 (*Bron et al., 1998*) were introduced to *B. subtilis* strains using natural competence (*Harwood and Cutting, 1990*). All *B. subtilis* strains used in this study are listed in *Table 2*, *E. coli* strains are listed in *Table 3*.

## Genetic screens

Genetic screens were performed as described previously (*Ho and Bradshaw, 2021*). Plasmids containing *rsbU* or *rsbT/rsbU*$^{Y28I}$ were subjected to PCR mutagenesis using GoTaq DNA polymerase mix (Promega) supplemented with 2 mM additional MgCl$_2$. Pools of mutagenized inserts were assembled into digested plasmid using isothermal (Gibson) assembly and transformed into *E. coli* DH5α cells. Mutation rate of approximately one single-nucleotide polymorphism per kilobase of DNA was confirmed by sequencing individual clones. Plasmids were then introduced to *B. subtilis* using natural competence and cells were plated on selective medium containing IPTG and X-gal for two days at 37 °C. Blue colonies were selected, restruck on plates with and without IPTG, and plasmids recovered from single colonies that retested as σ$^B$ positive were sequenced using Sanger sequencing. For the RsbU$^{Y28I}$ suppressor screen, two independent pools of PCR mutagenized plasmid were screened. In addition to the reported variants that were selected, we identified simple revertants that restored Y28 in both pools, and mutations in the promoter and RBS that we predict enhanced protein production. These variants were not studied further. Individual amino acid changing mutations were rebuilt in the parental plasmid using PCR (QuickChange) mutagenesis and the phenotypes were validated by replating on indicator plates. All strains shown in the figures are from these rebuilt plasmids.

## AlphaFold2 structure predictions

AlphaFold2 predictions were performed using ColabFold (*Mirdita et al., 2022*). Alphafold2_multimer_v2 was used in unpaired_paired mode with no templates with 3 recycles, 200 iterations, and greedy pairing strategy. The predicted aligned error plots for all AlphaFold2 structures are shown in *Figure 1—figure supplement 1*. The predicted local distance difference test scores (pLDDT) mapped onto the structures are shown in *Figure 1—figure supplement 2*.

**Table 3.** *E. coli* strains.

| Strain # | Genotype | Reference |
|---|---|---|
| RB45 | *BL21 (DE3) pET47b 6H-3C-rsbT* | *Ho and Bradshaw, 2021* |
| RB46 | *BL21 (DE3) pET47b 6H-3C-rsbU* | *Ho and Bradshaw, 2021* |
| RB47 | *BL21 (DE3) pET47b 6H-3C-rsbV* | *Ho and Bradshaw, 2021* |
| RB48 | *BL21 (DE3) pET47b 6H-3C-rsbW* | *Ho and Bradshaw, 2021* |
| RB49 | *BL21 (DE3) pET47b 6H-3C-rsbVW* | This study |
| RB50 | *BL21 (DE3) pET47b 6H-3C-rsbU$^{Q94L}$* | This study |
| RB51 | *BL21 (DE3) pET47b 6H-3C-rsbU$^{M166V}$* | *Ho and Bradshaw, 2021* |
| RB52 | *BL21 (DE3) pET47b 6H-3C-rsbTU* | This study |
| RB53 | *BL21 (DE3) pET47b 6H-3C-rsbU$^{R91E}$* | This study |
| RB54 | *BL21 (DE3) pET47b 6H-3C-rsbTU$^{Q94L}$* | This study |
| RB55 | *DH5 α Pspank-rsbU* | *Ho and Bradshaw, 2021* |
| RB56 | *DH5 α Pspank-rsbTU* | *Ho and Bradshaw, 2021* |
| RB57 | *DH5 α rsbU$^{Q94L}$* | |
| RB58 | *DH5 α Pspank-rsbU$^{M166V}$* | *Ho and Bradshaw, 2021* |
| RB59 | *DH5 α Pspank-rsbTU$^{Y28I}$* | This study |
| RB60 | *DH5 α Pspank-rsbTU$^{Y28I,S49G}$* | This study |
| RB61 | *DH5 α Pspank-rsbTU$^{Y28I,K53E}$* | This study |
| RB62 | *DH5 α Pspank-rsbTU$^{Y28I,T89A}$* | This study |
| RB63 | *DH5 α Pspank-rsbTU$^{Y28I,G92R}$* | This study |
| RB64 | *DH5 α Pspank-rsbTU$^{Y28I,G92E}$* | This study |
| RB65 | *DH5 α Pspank-rsbTU$^{Y28I,Q94L}$* | This study |
| RB66 | *DH5 α Pspank-rsbTU$^{Y28I,T110S}$* | This study |
| RB67 | *DH5 α Pspank-rsbTU$^{Y28I,V142A}$* | This study |
| RB68 | *DH5 α Pspank-rsbTU$^{Y28I,K143E}$* | This study |
| RB69 | *DH5 α Pspank-rsbTU$^{Y28I,C165Y}$* | This study |
| RB70 | *DH5 α Pspank-rsbTU$^{Y28I,M173I}$* | This study |
| RB71 | *DH5 α Pspank-rsbTU$^{R91E}$* | This study |
| RB72 | *DH5 α Pspank-rsbT$^{D92K}$U* | This study |
| RB73 | *DH5 α Pspank-rsbT$^{D92K}$U$^{R91E}$* | This study |
| RB74 | *DH5 α Pspank-rsbU$^{Q94I}$* | This study |
| RB75 | *DH5 α Pspank-rsbU$^{Q94V}$* | This study |
| RB76 | *DH5 α Pspank-rsbU$^{Q94Y}$* | This study |
| RB77 | *DH5 α Pspank-rsbU$^{Q94Y}$* | This study |
| RB78 | *DH5 α Pspank-rsbU$^{Q94F}$* | This study |
| RB79 | *DH5 α Pspank-rsbU$^{Q94W}$* | This study |
| RB80 | *DH5 α Pspank-rsbU$^{Q94M}$* | This study |
| RB81 | *DH5 α Pspank-rsbU$^{Q94N}$* | This study |
| RB82 | *DH5 α Pspank-rsbU$^{Q94E}$* | This study |
| RB83 | *DH5 α Pspank-rsbU$^{Q94A}$* | This study |

*Table 3 continued on next page*

*Table 3 continued*

| Strain # | Genotype | Reference |
|----------|----------|-----------|
| RB84 | *DH5 α Pspank-rsbU*$^{Q94G}$ | This study |
| RB85 | *DH5 α Pspank-rsbU*$^{T89A}$ | This study |
| RB86 | *DH5 α Pspank-rsbU*$^{C165Y}$ | This study |

## Socket2 analysis of AlphaFold2 structure predictions

Socket2 coiled-coil predictions were performed using Socket2 Colab (*Kumar and Woolfson, 2021*). Socket2 was used with a packing cutoff of 7.0 Å. Analysis done using Socket2 is shown in *Figure 5—figure supplement 2*.

## Materials availability statement

All strains, plasmids, and coordinate files reported in this manuscript are available by request to the corresponding author.

## Acknowledgements

The authors thank Richard Losick, Dorothee Kern, Chris Miller, Julia Kardon, Matthew Cabeen, Liz Hedstrom, Susan Lovett, Emily Stadnicki, and Maria-Eirini Pandelia for input at various stages of this project. Justin Curran, LuYing Pan, Spencer Clark, Alexis Ryan, and Wendy Yang additionally contributed to the research. We also thank students from University at Albany, SUNY who provided helpful comments as a review of our preprint. This research was supported by startup funds to NB from Brandeis University. RB was supported by T32 GM135126 and SP was supported by T32 GM007122. This research used resources of the Advanced Photon Source, a U.S. Department of Energy (DOE) Office of Science User Facility operated for the DOE Office of Science by Argonne National Laboratory under Contract No. DE-AC02-06CH11357. BioCAT was supported by grant P30 GM138395 from the National Institute of General Medical Sciences of the National Institutes of Health. Use of the Pilatus 3 1 M detector was provided by grant 1S10OD018090 from NIGMS. The content is solely the responsibility of the authors and does not necessarily reflect the official views of the National Institute of General Medical Sciences or the National Institutes of Health.

## Additional information

### Funding

| Funder | Grant reference number | Author |
|--------|------------------------|--------|
| Brandeis University | Startup Funds | Niels Bradshaw |
| National Institutes of Health | T32 GM135126 | Rishika Baral |
| National Institutes of Health | T32 GM007122 | Suhaily Caban-Penix |
| National Institutes of Health | P30 GM138395 | Jesse B Hopkins Maxwell B Watkins |

The funders had no role in study design, data collection and interpretation, or the decision to submit the work for publication.

### Author contributions

Rishika Baral, Conceptualization, Investigation, Visualization, Writing – original draft, Writing – review and editing; Kristin Ho, Conceptualization, Investigation, Writing – review and editing; Ramasamy P Kumar, Conceptualization, Formal analysis, Investigation; Jesse B Hopkins, Formal analysis, Investigation, Writing – review and editing; Maxwell B Watkins, Formal analysis, Investigation; Salvatore

LaRussa, Logan A Calderone, Investigation; Suhaily Caban-Penix, Resources, Investigation; Niels Bradshaw, Conceptualization, Supervision, Visualization, Writing – original draft, Writing – review and editing

**Author ORCIDs**
Rishika Baral ⓘ https://orcid.org/0000-0002-9365-8877
Ramasamy P Kumar ⓘ https://orcid.org/0000-0002-6555-8289
Jesse B Hopkins ⓘ https://orcid.org/0000-0001-8554-8072
Maxwell B Watkins ⓘ https://orcid.org/0000-0003-4559-2049
Suhaily Caban-Penix ⓘ https://orcid.org/0009-0007-6512-5399
Logan A Calderone ⓘ https://orcid.org/0000-0002-0960-3831
Niels Bradshaw ⓘ https://orcid.org/0000-0002-6845-4717

Reviewer #1 (Public review): https://doi.org/10.7554/eLife.100376.3.sa1
Reviewer #2 (Public review): https://doi.org/10.7554/eLife.100376.3.sa2
Reviewer #3 (Public review): https://doi.org/10.7554/eLife.100376.3.sa3
Author response https://doi.org/10.7554/eLife.100376.3.sa4

## Additional files

### Supplementary files
MDAR checklist

### Data availability
SAXS data were deposited in SASDB under accession codes SASDU85 and SASDU95. All other data generated or analyzed during this study are included in the manuscript and supporting files.

The following datasets were generated:

| Author(s) | Year | Dataset title | Dataset URL | Database and Identifier |
|---|---|---|---|---|
| Baral R, Kumar RP, Hopkins JB, Watkins MB, Bradshaw N | 2025 | Wild-type phosphoserine phosphatase RsbU dimer | https://www.sasbdb.org/data/SASDU85/ | SASDB, SASDU85 |
| Baral R, Kumar RP, Hopkins JB, Watkins MB, Bradshaw N | 2025 | Hyperactive variant of phosphoserine phosphatase RsbU (Q94L) bound to the activator serine/threonine-protein kinase RsbT (heterotetrameric complex) | https://www.sasbdb.org/data/SASDU95/ | SASDB, SASDU95 |

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
