## [Editor Report · eLife Assessment]

This **important** study combines genetic analysis, biochemistry, and structural modeling to reveal new insights into how changes in protein-protein structure activate signal transduction as part of the bacterial general stress response. The data, which was collected using validated and standard methods, and its interpretations are **convincing**; however, to fully meet the title's promise, additional experimental evidence is needed to strengthen the proposed model and its potential application to other systems. This manuscript will be of broad interest to microbiologists, structural biologists, and cell biologists.

---

## [Referee Report · Reviewer #1 (Public review)]

Summary:

This very interesting manuscript proposes a general mechanism for how activating signaling proteins respond to species specific signals arising from a variety of stresses. In brief, the authors propose that the activating signal alters the structure by a universal allosteric mechanism.

Strengths:

The unitary mechanism proposed is appealing and testable. The propose that the allosteric module consists of crossed alpha-helical linkers with similar architecture and that their attached regulatory domains connect to phosphatases or other molecules through coiled-coli domains, such that the signal is transduced via rigidifying the alpha helices, permitting downstream enzymatic activity. The authors present genetic and structural prediction data in favor of the model for the system they are studying, and stronger structural data in other systems.

Weaknesses:

I thank the authors for making significant revisions that addressed almost all of my concerns. I hope that the authors will consider addressing my last concern, which is that the title is inappropriate. However, I do not believe that this should hold up the publication of the ms.

"A General Mechanism for Initiating the General Stress Response in Bacteria" is misleading because it suggests a broadly applicable, universal mechanism across all bacterial species, whereas the study primarily focuses on *Bacillus subtilis* and its RsbU phosphatase activation. While the authors propose that the mechanism may extend to other bacteria, the evidence is largely based on structural modeling rather than direct experimental validation across multiple phyla. Additionally, the phrase "General Stress Response" might imply that the paper broadly explains stress response regulation, but it specifically examines the activation of RsbU by RsbT, which is just one really small part of the broader GSR network. The redundancy in "A General Mechanism for the General Stress Response" could also create an impression of an oversimplified, universal model when stress responses are often species- and context-specific. Furthermore, the study builds upon existing knowledge of partner-switching mechanisms rather than introducing an entirely new concept, making the claim of a general mechanism overstated and misleading for the field.

Title options could be "A Conserved Activation Mechanism for the General Stress Response Phosphatase in Bacteria", "Coiled-Coil Linker-Mediated Activation of a General Stress Response Phosphatase", all of which more accurately reflect the study's scope and findings.

---

## [Referee Report · Reviewer #2 (Public review)]

Summary:

While bacteria have the ability to induce genes in response to specific stresses, they also use the General Stress Response (GSR) to deal with growth conditions that presumably include a larger range of stresses (for instance, stationary phase growth). The activation of GSR-specific sigma factors is frequently at the heart of the induction of a GSR. Given the range of stresses that can lead to GSR induction, the regulatory inputs are frequently complex. In *B. subtilis*, the stressosome, a multi-protein complex, contains a set of proteins that, upon appropriate stresses, initiate partner switching cascades that free the sigma B sigma factor from an anti-sigma. The focus here is on the mode of activation of RsbU, a serine/threonine phosphatase of the PPM family, leading to sigB activation. RbsT, a component of the degradosome interacts with RsbU upon stress, activating the phosphatase activity. Once active, RsbU dephosphorylates its target (RsbV, an anti-antisigma), which in turn binds the anti-sigma. The conclusion is that flexible linker domains upstream of the phosphatase domain are the target for activation, resulting in a crossed-linker dimeric structure. The authors then use the information on RsbU to suggest that parallel approaches may be used to activate PPM phosphatases for the GSR response in other bacteria.

Strengths and Weaknesses:

(1) A strength of the work is the combination of modeling, genetics and biochemical approaches to support the idea that the flexibility of the linker of the RsbU phosphatase is critical to signalling and that this changes as a result of interactions of the signaling protein RsbT.

(2) The impact of the work, beyond better understanding of this particular signalling system, lies in the suggested parallels with other GSR system regulators in a range of bacteria. The work here provides fairly clear indications of what mutational changes would be most likely to test the model.

(3) Assuming that these predictions are shown to be correct in future work, that will leave as an intriguing question why this particular geometry has been conserved in GSR - whether they emerge from a common ancestor (found where?) and/or there is some characteristic (flexibility of modulating the response?) that is particularly important for GSR signal input. Coupled with this will be further understanding of how the linker and/or interacting proteins change in different systems.

---

## [Referee Report · Reviewer #3 (Public review)]

Summary:

The authors present a study building on their previous work on activation of the general stress response phosphatase, RsbU, from *Bacillus subtilis*. Using computed structural models of the RsbU dimer the authors map previously identified activating mutations onto the structure and suggest further protein variants to test the role of the predicted linker helix and the interaction with RsbT on the activation of the phosphatase activity.

Using in vivo and in vitro activity assays, the authors demonstrate that linker variants can constitutively activate RsbU and increase the affinity of the protein for RsbT, thus showing a link between the structure of the linker region and RsbT binding.

Small angle X-ray scattering experiments on RsbU variants alone, and in complex with RsbT show structural changes consistent with a decreased flexibility of the RsbU protein, which are hypothesised to indicate an disorder-order transition in the linker when RsbT binds. This interpretation of the data is consistent with the biochemical data presented by the authors.

Further computed structure models are presented for other protein phosphates from different bacterial species and the authors propose a model for phosphatase activation by partner binding. They compare this to the activation mechanisms proposed for histidine kinase two-component systems and GGDEF proteins and suggest the individual domains could be swapped to give a toolkit of modular parts for bacterial signalling.

Strengths:

The key mutagenesis data is presented with two lines of evidence to demonstrate RsbU activation - in vivo sigma-b activation assays utilising a beta-galactosidase reporter and in vitro activity assays against the RsbV protein, which is the downstream target of RsbU. These data support the hypothesis for RsbT binding to the RsbU linker region as well as the dimerisation domain to activate the RsbU activity.

Weaknesses:

Small angle scattering curves are difficult to unambiguously interpret, but the authors present good interpretations that fit with the biochemical data presented. These interpretations should be considered as models for future testing with other methods - hydrogen/deuterium exchange mass spectrometry, would be a good additional method to use, as exchange rates in the linker region would be affected significantly by the disorder/order transition on RsbT binding.

The interpretation of the computed structure models is provided with a few caveats related to the bias in the models returned by AlphaFold2. For the full-length models of RsbU and other phosphatase proteins, the relationship of the domains to each other is likely to be the least reliable part of the models - this is apparent from the PAE plots shown in supplementary figure 8.

Comments on revisions:

The authors have addressed the review comments satisfactorily for this manuscript to stand as a version of record.

---

## [Author Response]

The following is the authors’ response to the original reviews

**Public reviews:**

**Reviewer #1 (Public review):**
Summary:This very interesting manuscript proposes a general mechanism for how activating signaling proteins respond to species-specific signals arising from a variety of stresses. In brief, the authors propose that the activating signal alters the structure by a universal allosteric mechanism.Strengths:The unitary mechanism proposed is appealing and testable. They propose that the allosteric module consists of crossed alpha-helical linkers with similar architecture and that their attached regulatory domains connect to phosphatases or other molecules through coiled-coli domains, such that the signal is transduced via rigidifying the alpha helices, permitting downstream enzymatic activity. The authors present genetic and structural prediction data in favor of the model for the system they are studying, and stronger structural data in other systems.Weaknesses:The evidence is indirect - targeted mutations, structural predictions, and biochemical data. Therefore, these important generalizable conclusions are not buttressed by impeccable data, which would require doing actual structures in *B. subtilis*, confirming experiments in other organisms, and possibly co-evolutionary coupling. In the absence of such data, it is not possible to rule out variant models.

We thank the reviewer for their feedback. A challenge of studying flexible proteins is that it is often not possible to directly obtain high resolution structural data. For the case of *B. subtilis* RsbU, the independent experimental approaches we applied (including two unbiased genetic screens, targeted mutagenesis, SAXS, enzymology, and structure prediction, which includes evolutionary coupling) converged upon a model for activation, which we feel is well supported. Frustratingly, our attempts at determining high resolution experimental structures have been unsuccessful, which we think is due to the flexibility of the proteins revealed by our SAXS experiments. For example, we collected X-ray diffraction data from crystals of a fragment of *B. subtilis* RsbU containing the N-terminal domain and linker in which the linker was almost entirely disordered in the maps. We agree that doing experiments in other organisms would be valuable next steps to test the hypothesis that this coiled-coil based transduction mechanism is conserved across species, and will modify the text to differentiate this more speculative section of the manuscript.

We have modified the abstract to read:

“This coiled-coil linker transduction mechanism additionally suggests a resolution to the mystery of how shared sensory domains control serine/threonine phosphatases, diguanylate cyclases and histidine kinases.”

We have modified the results to read:

"These predictions suggest a testable hypothesis that RsbP is controlled through an activation mechanism similar to that of RsbU (Fig. 5A)”

“From this analysis, we speculate that linker-mediated phosphatase domain dimerization is an evolutionarily conserved, adaptable mechanism to control PPM phosphatase activity.”

Based on this critique (and the critiques of the other reviewers), we plan to do energetic analysis of the predicted coiled coils from the enzymes we analyzed from other species and to incorporate this into the manuscript.

We have modified the results to read:

Consistent with a model in which the stability of the linker plays a conserved regulatory role, the AlphaFold2 models for many of the predicted structures have unfavorable polar residues buried in the coiled-coil interface (positions a and d, for which non-polar residues are most favorable) (Figure 5 – figure supplement 2).”

Finally, in the manuscript, we have highlighted that this mechanism is not the only mechanism for activation of other proteins with effector domains connected to linkers, but rather one of many mechanisms (Fig 5G). The reviewer additionally made helpful suggestions about the text in detailed comments that we will incorporate as appropriate.

**Reviewer #2 (Public review):**
Summary:While bacteria have the ability to induce genes in response to specific stresses, they also use the General Stress Response (GSR) to deal with growth conditions that presumably include a larger range of stresses (for instance, stationary phase growth). The activation of GSR-specific sigma factors is frequently at the heart of the induction of a GSR. Given the range of stresses that can lead to GSR induction, the regulatory inputs are frequently complex. In *B. subtilis*, the stressosome, a multi-protein complex, contains a set of proteins that, upon appropriate stresses, initiate partner switching cascades that free the sigma B sigma factor from an anti-sigma. The focus here is on the mode of activation of RsbU, a serine/threonine phosphatase of the PPM family, leading to sigB activation. RbsT, a component of the degradosome interacts with RsbU upon stress, activating the phosphatase activity. Once active, RsbU dephosphorylates its target (RsbV, an anti-antisigma), which in turn binds the anti-sigma. The conclusion is that flexible linker domains upstream of the phosphatase domain are the target for activation, via binding of proteins to the N-terminal domain, resulting in a crossed-linker dimeric structure. The authors then use the information on RsbU to suggest that parallel approaches are used to activate PPM phosphatases for the GSR response in other bacteria. (Biology vs. Mechanism, evolution?)Strengths and Weaknesses:Many of these have to do with clarifying what was done and why. This includes the presentation and content of the figures.One issue relates to the background and context. A bit more information on the stresses that release RsbT would be useful here. The authors might also consider a figure showing the major conclusions and parallels for SpoIIE activation and possibly other partner switches that are discussed, introducing the switch change more clearly to set the stage for the work here (and the generalization). There are a lot of players to keep track of.

We plan to carefully review the manuscript to improve the clarity of presentation and background. In particular, we thank the reviewer for pointing out the missing information about the release of RsbT from the stressosome. We will incorporate this information into the introduction and provide an additional figure.

We have added the following text to the introduction:

“RsbT is sequestered in a megadalton stress sensing complex called the stressosome, and is released to bind RsbU in response to specific stress signals including ethanol, heat, acid, salt, and blue light”

We have added a new figure panel (2C) that shows the model for how Q94L, M166V, and RsbT binding induce conformational change of the PPM domain to recruit metal cofactor and activate RsbU (analogous, but slightly different from the mechanism for SpoIIE).

The reviewer additionally provided detailed helpful comments that we will incorporate in the text and figures.

**Reviewer #3 (Public review):**
Summary:The authors present a study building on their previous work on activation of the general stress response phosphatase, RsbU, from *Bacillus subtilis*. Using computed structural models of the RsbU dimer the authors map previously identified activating mutations onto the structure and suggest further protein variants to test the role of the predicted linker helix and the interaction with RsbT on the activation of the phosphatase activity.Using in vivo and in vitro activity assays, the authors demonstrate that linker variants can constitutively activate RsbU and increase the affinity of the protein for RsbT, thus showing a link between the structure of the linker region and RsbT binding.Small angle X-ray scattering experiments on RsbU variants alone, and in complex with RsbT show structural changes consistent with a decreased flexibility of the RsbU protein, which is hypothesised to indicate a disorder-order transition in the linker when RsbT binds. This interpretation of the data is consistent with the biochemical data presented by the authors.Further computed structure models are presented for other protein phosphates from different bacterial species and the authors propose a model for phosphatase activation by partner binding. They compare this to the activation mechanisms proposed for histidine kinase two-component systems and GGDEF proteins and suggest the individual domains could be swapped to give a toolkit of modular parts for bacterial signalling.Strengths:The key mutagenesis data is presented with two lines of evidence to demonstrate RsbU activation - in vivo sigma-b activation assays utilising a beta-galactosidase reporter and in vitro activity assays against the RsbV protein, which is the downstream target of RsbU. These data support the hypothesis for RsbT binding to the RsbU linker region as well as the dimerisation domain to activate the RsbU activity.Weaknesses:Small angle scattering curves are difficult to unambiguously interpret, but the authors present reasonable interpretations that fit with the biochemical data presented. These interpretations should be considered as good models for future testing with other methods - hydrogen/deuterium exchange mass spectrometry, would be a good additional method to use, as exchange rates in the linker region would be affected significantly by the disorder/order transition on RsbT binding.

We agree with the reviewer that the SAXS data has inherent ambiguity due to the nature of the measurement. However, SAXS is one of the best techniques to directly assess conformational flexibility. Our scattering data for RsbU have multiple signatures of flexibility supporting a high confidence conclusion. While the scattering data support a reduction in flexibility for the RsbT/RsbU complex, we agree that a high resolution structure would be valuable. However the combination of the scattering data with our biochemical and genetic data supports the validity of the AlphaFold predicted model. We thank the reviewer for the suggestion of future hydrogen/deuterium exchange experiments that would be complementary, but which we feel are beyond the scope of this work.

The interpretation of the computed structure models should be toned down with the addition of a few caveats related to the bias in the models returned by AlphaFold2. For the full-length models of RsbU and other phosphatase proteins, the relationship of the domains to each other is likely to be the least reliable part of the models - this is apparent from the PAE plots shown in Supplementary Figure 8. Furthermore, the authors should show models coloured by pLDDT scores in an additional supplementary figure to help the reader interpret the confidence level of the predicted structures.

We thank the reviewer for suggestions on how to clarify the discussion of AlphaFold models. We will decrease the emphasis on the computed models in the text and will add figures with the models colored by the pLDDT scores to aid in the interpretation.

We have modified the text of the Abstract: “This coiled-coil linker transduction mechanism additionally suggests a resolution to the mystery of how shared sensory domains control serine/threonine phosphatases, diguanylate cyclases and histidine kinases.”

We have modified the text of the Results: “These predictions suggest a testable hypothesis that RsbP is controlled through an activation mechanism similar to that of RsbU (Fig. 5A).”

“From this analysis, we speculate that linker-mediated phosphatase domain dimerization is an evolutionarily conserved, adaptable mechanism to control PPM phosphatase activity”

We have also added Figure 1 – figure supplement 2 with the AlphaFold2 models colored by the pLDDT scores.

**Recommendations for the authors:**

**Reviewer #1 (Recommendations for the authors):**
Baral and colleagues investigate the regulatory mechanisms of the General Stress Response (GSR) in *Bacillus subtilis*, focusing on the phosphatase RsbU and its regulation by the protein RsbT. The GSR is a critical adaptive mechanism that allows bacteria to survive under various stress conditions by reshaping their physiology through a broad transcriptional response. RsbU, a key player in the GSR, facilitates the activation of the transcription factor SigB by dephosphorylating RsbV. This activation is mediated through a partner-switching mechanism involving RsbT. Baral and colleagues use a combination of genetic screening, structural predictions via AlphaFold2, and biophysical techniques such as SAXS and MALS to present a model for how RsbT regulates RsbU. Key findings include the identification of specific amino acid substitutions that enhance RsbU activity, the role of the α-helical linker in RsbU dimerization and activation, and the potential broader conservation of these mechanisms across bacterial species. However, as described below, additional work is required to solidify the results.Major Points(1) The manuscript is misnamed--it dissects a single step of the signal-transduction pathway regulating the general stress response. Instead, it is rather seeking a generalizable mechanism for kinase -phosphatase interactions across stresses.

We have edited the title to “A General Mechanism for Initiating the General Stress Response in Bacteria” to reflect that that this study addresses the initiating event of the general stress response.

(2) The genetic screen likely has limitations in detecting all possible variants that could affect RsbU activity. The readout is specific to σ^B activation, and the focus on specific amino acid substitutions may overlook other significant regions or mechanisms involved in the regulation of RsbU, particularly those involving RsbV and RsbT.

Our screens were specifically designed to identify features of RsbU that contribute to regulation. Importantly, RsbU does not have any known targets other than RsbV and the downstream σ^B^ response but agree that substitutions in either RsbV or RsbT could influence RsbU activation. In principle our suppressor screen with RsbU^Y28I^ could have identified RsbT variants (*rsbT* was mutagenized in this screen), but we did not identify any such variants in the screen. We conducted a separate screen (published elsewhere) that specifically addressed how RsbU recognizes RsbV.

(3) The authors largely focus on the biochemical and structural aspects of RsbU regulation. There is limited discussion on the broader functional implications of these findings in the context of bacterial physiology and stress response. Incorporating more in vivo studies to show how these mechanisms impact bacterial survival and adaptation would provide a more comprehensive understanding.

We appreciate this comment, but did not conduct additional studies of survival and adaptation because the phenotypes of σ^B^ deletion in *B. subtilis* under laboratory conditions are relatively mild and therefore difficult to assay. Future studies to address this in other systems could be highly informative.

(4) The results primarily support the model of linker-mediated dimerization and rigidity. However, other potential regulatory mechanisms or interacting partners might also play significant roles in RsbU activation. A more thorough exploration of these possibilities would strengthen the study's conclusions.

One of the major advantages of RsbU as a model for initiation of the general stress response is that the system is discreet with all evidence pointing to there being a single primary input (RsbT) and output (dephosphorylation of RsbV). While there are other possible variations on the system (for example RsbU may be directly activated by manganese stress), we focused on this system precisely because of its simplicity.

(5) While the study presents evidence for the conservation of the described mechanism across different species, this assumption is based on structural predictions and limited experimental data. Broader experimental validation across diverse bacterial species would be necessary to substantiate this claim. Coevolution coupling along with conservation/evolutionary studies could be considered.

We have altered the language in the paper to emphasize where we are making inferences from predictions that are therefore more speculative. We agree that a more detailed analysis of the evolutionary coupling would likely be fruitful. We note that these couplings are the major driving force of AlphaFold predictions, suggesting that these couplings contributed to the models that we analyzed.

(6) The reliance on AlphaFold2 for structural predictions introduces potential biases and uncertainties inherent in computational models. Experimental validation of these models through additional techniques such as cryo-EM or X-ray crystallography would strengthen the conclusions.

We agree with this point, which is why we performed extensive analysis and validation of the models for RsbU using SAXS, genetics, and biochemistry. The proposed techniques are made more challenging by flexibility and heterogeneity, which we detected in our experiments. Our attempts thus far at experimental structure determination are consistent with this being a major technical hurdle.

(7) SAXS data provide low-resolution structural information, and the interpretation of flexibility versus rigidification might be overemphasized in its interpretation. This part of the study was difficult to interpret. Improving readability by breaking down the text into sections with clear headings for each figure panel and clarifying descriptions of the panels and methods would help. Complementary high-resolution techniques could provide a more definitive view of the linker's conformational changes.

We have modified the presentation of the figures to clarify the SAXS analysis. The fact that the SAXS analysis suggests flexibility rather than a discrete inactive conformation means that high-resolution techniques may not be appropriate for this system.

(8) The study primarily focuses on the model where RsbT binding rigidifies the RsbU linker. Alternative hypotheses, such as subtle conformational adjustments without complete rigidification, are not extensively explored or ruled out.

Our analysis of the SAXS data strongly suggests that a subtle conformational change could not account for the scattering data that we obtained. We have modified the text to clarify this point.

“Indicative of significant deviation between the RsbU structure in solution to the AlphaFold2 model, the scattering intensity profile (I(q) vs. q) was a poor fit (χ^2^ 12.53) to a profile calculated from the AlphaFold2 model of an RsbU dimer using FoXS (Schneidman-Duhovny et al. 2016; Schneidman-Duhovny et al. 2013) (Fig. 4A). We therefore assessed the SAXS data for the RsbU dimer for features that report on flexibility (Kikhney & Svergun 2015). First, the scattering intensity data lacked distinct features caused by the multi-domain structure of RsbU from the AlphaFold2 model (Fig.4A).”

(9) Future studies should aim to validate the AlphaFold2 predictions with high-resolution structural techniques. This would provide definitive evidence for the proposed conformational states of RsbU with and without RsbT.

The fact that the SAXS analysis suggests flexibility rather than a discrete inactive conformation means that high-resolution techniques may not be appropriate for this system.

(10) Investigating the RsbU-RsbT interaction in vivo using techniques like FRET, co-immunoprecipitation, or live-cell imaging would provide a more comprehensive understanding of their functional dynamics in a cellular context.

We appreciate the reviewer’s suggestions for future experiments.

(11) Exploring and testing alternative models of RsbU activation, such as partial rigidification or different modes of conformational change, would strengthen the conclusions.

While our data strongly support that a flexible-to-rigid transition controls RsbU activation, we agree that it is possible that other mechanisms of linker modification could control other phosphatases and we discuss this at some length in the discussion.

(12) The figure legends are quite dense and could benefit from some streamlining.

We have edited the figure legends for clarity and length.

**Reviewer #2 (Recommendations for the authors):**
(1) Activation assays (Figures 1, 3, S2) are presented here as blue or white spots (reflecting a reporter activity). While off and on these are fairly clear, it is more difficult to compare the degree of activity (for instance that *rsbUQ94L* is more active than M166V). It would also be good to clearly present in the text the logic of asking if the mutant is RsbT independent or not (and the interpretation of that). Quantitative assays of these would be very useful.

We chose not to perform quantitative-LacZ assays here because of several complications to interpreting these results that we encountered in our previously published study (Ho and Bradshaw, 2021). However, the level of blue pigmentation shown in Figure 1B for RsbU Q94L and RsbU M166V is qualitatively different, making the comparison possible. Most importantly, we observed cell density dependent changes in LacZ activity in the absence of *rsbT* for *rsbUM166V* expressing cells, meaning that comparisons between strains would be difficult. Additionally, we found that it was important to make a chromosomal replacement of *rsbU* to see the full effect of the M166V substitution. However, we were not able to construct a similar *rsbUQ94L* strain, likely because the high level σ^B^ activity is lethal (we were able to construct this strain when σ^B^ was deleted but only obtained strains with additional loss-of-function mutations in RsbU when σ^B^ was present).

We have modified the text to explain the logic of identifying RsbT independent variants: “We previously conducted a genetic screen (Ho & Bradshaw 2021) to identify features of RsbU that are important for phosphatase regulation by isolating gain-of-function variants that are active in the absence of RsbT.”

(2) Explain Figure S8 graphs: as much as Alphafold is now in use, the authors should provide some further explanation of what is shown here. Blue (low error) is good, presumably. What are the A, B, C, and D sections showing? Different parts of a given letter region (and between them)? What is the x-axis? Is the top-ranked model used in every case in the text? How different are these models? The Methods section could be used for some of this (but doesn't in its current form). This also becomes important for the models generated later in the paper (Figure S7), which look rather different here.

We have modified figure S8 to include additional labels and have added structures with the pLDDT scores shown. We have additionally modified the figure legends and methods to provide the requested information.

(3) Figure 1C, D, Figure S2: amino acid ends of linker domains could be shown (text discusses 83-97 the linker as a two-turn coiled coil; Q94 is pretty close to the end of this coiled-coil? Figure S2 is even less clear - addresses of other amino acids would help, and or an added sequence showing the full linker and coiled-coil region). Some explanation for positions for readers to focus on for full coiled-coil would be useful in the legend of Figure S2. How strong a coiled-coil prediction is there for this region?

We have added the sequence of the coiled-coil regions to the figures with numbering. For these analyses we used the Socket2 program, which analyzes a PDB file to identify coiled-coil regions and thus does not provide a confidence score. However, inspection of the sequence and the confidence scores of the AlphaFold2 models indicates that the coiled-coil regions are not ideal, consistent with this being a regulatory feature.

Is it clear that the fully inactive proteins are still properly folded and soluble?

In the case of RsbU, our biophysical analysis indicates that the inactive form of the protein is soluble. While phosphatase activity is substantially reduced, our unpublished comparison of single- and multiple-turnover reactions in the absence of RsbT indicates that nearly all of the enzyme is active.

Finally, are there other positions that would also be expected, from this model, to stabilize the coiled-coil and thus bypass the requirement for RsbT? If so, it would be good to test these. Is it the burial of amino acid at position 94 that is important, or the ability to form crossed helices?

Because of how short the predicted coiled-coil region is, we did not identify any obvious positions that would likely have the same effect as Q94 substitution. We considered making helix-breaking mutations, which would be predicted to block RsbU activation, but favored analysis of the wildtype protein because of limitations in interpreting the effects of loss-of-function mutations.

(4) Figure 2A, RsbT binding to RsbU: It was not entirely clear to this reviewer why one would expect the RsbT binding, not needed for activation, to be increased by the mutation that stabilizes the crossed alpha helices. The change is impressive but doesn't the lack of a need for RsbT suggest that this mutation bypasses the normal mechanism? (Is dimerization enuf? Or other protein cross helices?).

We have modified the text to clarify this point: “One prediction of our hypothesis that RsbT stabilizes the crossed alpha helices of the RsbU dimer, is that RsbT should bind more tightly to *rsbUQ94L* than to *RsbU* because the coiled-coil conformation that RsbT binds would be more energetically favorable.” Another way of putting this is that if the Q94L substitution activates RsbU through an on-pathway mechanism, RsbT *must* bind more tightly.

(5) Figure 3A, Figure S3: Please label the yellow (interface) residues in RsbU and RsbT in Fig. S3 and the green (suppressor) spheres in Figure 3A.

We have added labels to the figures as suggested.

If RbsT interacts with the N-terminal dimerization domain and linker, why were residues 174 and 178 (from PPM domain) shown to be implicated in binding?

The fact that residues in the switch region suppress a mutation that decreases RsbT binding suggests that this region is part of an allosteric network that links RsbT binding, the linker, and dimerization of the phosphatase domains. For example, any substitution that promotes a conformation of the phosphatase domain that is more favorable for dimerization would also promote RsbT binding. However, the precise details of how each mutation fits into this network is not clear and we have therefore chosen to not specify a particular model to avoid over interpreting our data.

Are these marked in Figure S3?

We have added labels to make this clear.

Are these part of a dimerization interface in the C-terminal domain? Are any/all of these RsbU mutants suppressed by Q94L, as one might predict (apparently Y28I is since Q94L was again identified)?

We chose to focus on Y28I because it was the best studied previously, but we would predict that Q94L would suppress other RsbT binding mutations.

(6) Line 191-192: Is it surprising that no suppressors were isolated in RsbT?

We didn’t have a preconception of whether or not it would be possible to identify similar suppressors in RsbT. Explanations for why we did not identify such suppressors could include that RsbT may be destabilized more easily by substitution, that RsbT is more constrained because it has other interaction partners, or that the particular substitutions that would suppress Y28I are less common by the PCR mutagenesis strategy we used.

(7) Figure 3: Would the same mutants arise if the screen had been done in the absence of RsbT? Was RsbT-dependent tested for the rsbU alleles?

Our prediction is that we would not have identified any of these mutations except for Q94L in the absence of *rsbT.* We tested a few of the alleles and found them all to be *rsbT* dependent, but did not systematically test all of the alleles and therefore did not include this analysis in the manuscript.

Given the findings earlier in the paper for Q94L, suggesting that this stabilizes the coiled-coil and shows some activity in the absence of RsbT, it seems that the interpretation of other mutants in this region (and Q94L itself) as evidence that RsbT contacts the linker directly and that contact is necessary for activation may be an overinterpretation. If these are in fact RsbT independent, they support the importance of the linker (do they further stabilize coiled-coil formation?), rather than the role of RsbT here. Are G92 and T89 on the outside of the coiled-coil? If Q94 is buried, is it qualitatively different from these others?

G92 and T89 are predicted to be exposed. The fact that these mutations are near Q94 is part of the reason that we focused on R91 and the predicted contact with D92 of RsbT as another approach to validate the predicted interface.

(8) Figure 3C addresses the issue of direct interaction of RsbT with the RsbU linker to some extent, given that RsbU R91E doesn't appear to have a lot of activity without RsbT. It would be helped by telling the reader what the R91 contact is initially.

We have modified the text to clarify this point: “To test the model that RsbT activates RsbU by directly interacting with the linker to dimerize the RsbU phosphatase domains, we introduced a charge swap at position R91 that would abolish a predicted salt-bridge with RsbT D92 (Fig. 3C).”

(9) Figure 4 and the discussion of it in the text is not likely to be easily understandable for many readers. Aside from providing a bit more explanation of what these analyses are showing, it would be useful to start the whole section (or maybe even much earlier in the paper) with the information found on lines 261-264, that other studies show that the N-terminus dimerizes stably on its own (and is it known that the C-terminus does not?). Then the discussion of the alternative models early in this section would be clearer.

We have updated the introduction to emphasize this point “RsbU has an N-terminal four-helix bundle domain that dimerizes RsbU and is also the binding site for RsbT, which activates RsbU as a phosphatase (Fig. 1C,D) (Delumeau et al. 2004).”

We have also added clarification to the model presented at the beginning of this section: “A second possibility is that inactive RsbU is dimerized by the N-terminal domains but that the linkers of inactive RsbU are flexible and that the phosphatase domains only interact with each other when RsbT orders the linkers into a crossing conformation.”

Is the dimerization of the N-terminal domains previously determined similar/the same as what is seen in the AlphaFold models used here (or the AlphaFold dimerization derived primarily from that data?).

Yes, the dimerization in the AlphaFold models matches closely to the published structure.

(10) Discussion and Figure 5: The final part of this work predicts AlphaFold models for a set of other phosphatases involved in initiating GSR across bacterial species, and suggests that linked-mediated phosphatase dimerization is the critical factor to activate the phosphatase. Clearly, this is the most speculative but interesting aspect of the paper. A number of possible questions are suggested by some of this:a. Do any of the activating mutants In RsbU and RsbP in the PPM domain (that apparently improve dimerization and thus activation) do a similar job in the other modeled proteins?

This is an interesting question, but unfortunately most of these proteins have not been biochemically characterized. We highlight examples of RsbP and *E. coli* RssB for which similar activating mutations have been characterized.

b. The legend (Figure 5G) suggests that all of the linker combinations will be coiled-coils, but that they will undergo different types of activating (and dimerizing?) transitions. Is that in fact what is being proposed here?

Yes, this is our working hypothesis.

c. If there is no dimerization (as noted, only weak dimerization has been reported for *E. coli* RssB), does that generalize the model to there are linkers and their structures are important? At the least, would the folding up of the *E. coli* RssB linker with antiadaptor binding be considered another mode of signal transduction or rather some sort of storage form?

Interestingly, the *P. aeruginosa* RssB constitutively dimerizes, suggesting the *E. coli* is the outlier.

d. Would the "toolkit" model, in which different changes occur in the linker regions, suggest that the interacting proteins are going to be critical for the type of linker changes that will be important? Or something about the nature of the linkers themselves?

This is an interesting question that we cannot yet answer. We have chosen to focus on the possible flexibility of this mechanism and anticipate that a variety of mechanisms will be used.

e. Given the extensive comparison to *E. coli* RssB, the authors might consider a figure to clarify the relative domain architecture, sequences that are akin to switch regions, and others important to the discussion here.

We tried to highlight this in Figure 5C including coloring the regions similar to the switch regions.

**Reviewer #3 (Recommendations for the authors):**
Given the caveats noted above related to the reliability of computed structure models, I would recommend the authors make the following additions/modifications to their manuscript:(1) The authors should show alpha fold models coloured by pLDDT scores in an additional supplementary figure to help the reader interpret the confidence level of the predicted structures.

We have added these models to figure 1 – figure supplement 2.

(2) Because of the points mentioned above the authors should tone down the generalisation relating to the activation mechanism of this family of phosphatases presented in the discussion.

We have modified the paper throughout to emphasize where we are speculating.